# Robustifying State-space Models for Long Sequences via Approximate Diagonalization

**Annan Yu,**[1]  **Arnur Nigmetov,**[2]  **Dmitriy Morozov,**[2]
**Michael W. Mahoney,**[2,3,4]  **N. Benjamin Erichson**[2,3]
[1] Center for Applied Mathematics, Cornell University, Ithaca, NY 14853, USA
[2] Lawrence Berkeley National Laboratory, Berkeley, CA 94720, USA
[3] International Computer Science Institute, Berkeley, CA 94704, USA
[4] Department of Statistics, University of California at Berkeley, Berkeley, CA 94720, USA
`ay262@cornell.edu`,  `{anigmetov,dmorozov}@lbl.gov`,
`mmahoney@stat.berkeley.edu`,  `erichson@icsi.berkeley.edu`

## Abstract

State-space models (SSMs) have recently emerged as a framework for learning long-range sequence tasks. An example is the structured state-space sequence (S4) layer, which uses the diagonal-plus-low-rank structure of the HiPPO initialization framework. However, the complicated structure of the S4 layer poses challenges; and, in an effort to address these challenges, models such as S4D and S5 have considered a purely diagonal structure. This choice simplifies the implementation, improves computational efficiency, and allows channel communication. However, diagonalizing the HiPPO framework is itself an ill-posed problem. In this paper, we propose a general solution for this and related ill-posed diagonalization problems in machine learning. We introduce a generic, backward-stable "perturb-then-diagonalize" (PTD) methodology, which is based on the pseudospectral theory of non-normal operators, and which may be interpreted as the approximate diagonalization of the non-normal matrices defining SSMs. Based on this, we introduce the S4-PTD and S5-PTD models. Through theoretical analysis of the transfer functions of different initialization schemes, we demonstrate that the S4-PTD/S5-PTD initialization strongly converges to the HiPPO framework, while the S4D/S5 initialization only achieves weak convergences. As a result, our new models show resilience to Fourier-mode noise-perturbed inputs, a crucial property not achieved by the S4D/S5 models. In addition to improved robustness, our S5-PTD model averages 87.6% accuracy on the Long-Range Arena benchmark, demonstrating that the PTD methodology helps to improve the accuracy of deep learning models.

## 1 Introduction

Sequential data are pervasive across a wide range of fields, including natural language processing, speech recognition, robotics and autonomous systems, as well as scientific machine learning and financial time-series analysis, among others. Given that many of these applications produce exceedingly long sequences, sequential models need to capture long-range temporal dependencies in order to yield accurate predictions. To this end, many specialized deep learning methods have been developed to deal with long sequences, including recurrent neural networks (RNNs) (Arjovsky et al., 2016; Chang et al., 2019; Erichson et al., 2021; Rusch & Mishra, 2021; Orvieto et al., 2023), convolutional neural networks (CNNs) (Bai et al., 2018; Romero et al., 2022), continuous-time models (CTMs) (Gu et al., 2021; Yildiz et al., 2021), and transformers (Katharopoulos et al., 2020; Choromanski et al., 2020; Kitaev et al., 2020; Zhou et al., 2022; Nie et al., 2023).

Over the past few years, the new class of state-space models (SSMs) gained vast popularity for sequential modeling due to their outstanding performance on the Long-Range Arena (LRA) dataset (Tay et al., 2021). An SSM is built upon a continuous-time linear time-invariant (LTI) dy-

namical system $\Sigma = (\mathbf{A}, \mathbf{B}, \mathbf{C}, \mathbf{D})$, which is a system of linear ODEs given by

$$\begin{aligned}
\mathbf{x}'(t) &= \mathbf{A}\mathbf{x}(t) + \mathbf{B}\mathbf{u}(t), \\
\mathbf{y}(t) &= \mathbf{C}\mathbf{x}(t) + \mathbf{D}\mathbf{u}(t),
\end{aligned} \tag{1}$$

where $\mathbf{A} \in \mathbb{C}^{n \times n}$, $\mathbf{B} \in \mathbb{C}^{n \times m}$, $\mathbf{C} \in \mathbb{C}^{p \times n}$, $\mathbf{D} \in \mathbb{C}^{p \times m}$ are the state, input, output and feedthrough matrices; and $\mathbf{u}(t) \in \mathbb{C}^m, \mathbf{x}(t) \in \mathbb{C}^n, \mathbf{y}(t) \in \mathbb{C}^p$ are the inputs, states, and outputs of the system, respectively. The system can be discretized at time steps $j\Delta t$, where $\Delta t > 0$ and $j = 1, \dots, L$, to be fed with sequential inputs of length $L$. To store and process the information of the long sequential inputs online, the SSMs are often initialized by a pre-designed LTI system. One of the most popular schemes is called "HiPPO initialization" (Voelker et al., 2019; Gu et al., 2020), in which the Legendre coefficients of the input history at time $t$, i.e., $\mathbf{u} \cdot \mathbb{1}_{[0,t]}$, are stored and updated in the state vector $\mathbf{x}(t)$. This initialization is specifically designed to model long-range dependencies in sequential data. The recently proposed S4 model (Gu et al., 2022b) leverages the HiPPO initialization and accelerates training and inference by decomposing $\mathbf{A}$ into the sum of a diagonal matrix and a low-rank one. The diagonal-plus-low-rank (DPLR) structure yields a barycentric representation (Antoulas & Anderson, 1986) of the transfer function of eq. (1) that maps inputs to outputs in the frequency domain, enabling fast computation in the frequency domain (Aumann & Gosea, 2023).

While the DPLR structure achieves an asymptotic speed-up of the model, considering $\mathbf{A}$ to be a diagonal matrix results in a simpler structure. Compared to a DPLR matrix $\mathbf{A}$, a diagonal SSM is not only faster to compute and easier to implement, but it also allows integrating channel communication via parallel scans (Smith et al., 2023), thereby improving its performance on long-range tasks. Unfortunately, the problem of diagonalizing the HiPPO framework is exponentially ill-conditioned, as $n$ increases. Hence, while Gu et al. (2022b) shows analytic forms of the eigenvalues and eigenvectors of HiPPO matrices, they suffer from an exponentially large variance and cannot be used in practice. So far, the most popular way of obtaining a diagonal SSM is to simply discard the low-rank part from the DPLR structure, leveraging a stable diagonalization algorithm for a normal matrix. Discarding the low-rank component changes the underlying diagonalization problem, however; and it abandons the theoretical insights about HiPPO. Still, the resulting model almost matches S4's performance, in practice. Such diagonal models are called S4D (Gu et al., 2022a) when the systems are single-input/single-output (i.e., $m = p = 1$) and S5 (Smith et al., 2023) when the systems are multiple-input/multiple-output (i.e., $m = p > 1$), which enables channel communication.

The issue of ill-posed diagonalization problems is not merely specific to SSMs. For example, it is known that non-normal matrices make RNNs more expressive (Kerg et al., 2019; Orhan & Pitkow, 2020). More generally, non-normality plays an important role in the training of certain neural networks (Sengupta & Friston, 2018; Kumar & Bouchard, 2022). While the ill-posedness of the diagonalization problem essentially prevents accurate computation of eigenvalues and eigenvectors (i.e., we cannot have a small forward error) — in fact, the true spectral information becomes meaningless[1] — using a backward stable eigensolver, one can recover the non-normal matrix accurately (i.e., we can have a small backward error) from the wrong eigenvalues and eigenvectors.

In this paper, we propose a generic "perturb-then-diagonalize" (PTD) methodology as a backward stable eigensolver. PTD is based on the idea that a small random perturbation remedies the problem of the blowing up of eigenvector condition number (Davies, 2008; Davies & Hager, 2009; Banks et al., 2021), regularizing the ill-posed problem into a close but well-posed one. It is based on the pseudospectral theory of non-normal operators (Trefethen & Embree, 2005)[2] and may be interpreted as the approximate diagonalization of the non-normal matrices.

Our PTD method can be used to diagonalize the highly non-normal HiPPO framework. Therefore, instead of using the eigenvalues of the normal component of the HiPPO matrix to initialize the matrix $\mathbf{A}$ as in the S4D and S5 models, we propose to initialize $\mathbf{A}$ using the eigenvalues of a perturbed HiPPO matrix (see section 4). The resulting S4-PTD and S5-PTD models are shown to be more robust than their S4D and S5 companions under certain Fourier-mode perturbations. Our method is flexible and can be used to diagonalize many SSM initialization schemes that may be invented in the future.

---

[1]If an eigenvector matrix $\mathbf{V}$ is ill-conditioned, then projecting a vector onto the eigenbasis is unstable so the eigendecomposition suffers from a large variance and does not reveal any useful information of the matrix.

[2]The pseudospectral theory studies the effect of perturbations on the spectrum of a non-normal operator.

**Contribution.** Here are our main contributions: **(1)** We propose a "perturb-then-diagonalize" (PTD) methodology that solves ill-posed diagonalization problems in machine learning when only the backward error is important. **(2)** We provide a fine-grained analysis that compares the S4 and the S4D initialization. In particular, we quantify the change of the transfer function when discarding the low-rank part of HiPPO, which is done in the diagonal S4D/S5 initialization. We show that while the outputs of the S4D/S5 system on a *fixed* smooth input converge to those of the S4 system at a linear rate as $n \to \infty$, the convergence is not uniform across all input functions (see section 3.1). **(3)** Based on our theoretical analysis, we observe, using the sequential CIFAR task (see section 5.2), that the S4D/S5 models are very sensitive to certain Fourier-mode input perturbations, which impairs the robustness of the models. **(4)** We propose the S4-PTD and S5-PTD models that replace the normal component of the HiPPO matrix, used to initialize the S4D and S5 models, with a perturbed HiPPO matrix. Our models are robust to Fourier-mode input perturbations. We theoretically estimate the effect of the perturbation (see section 4). We propose computing the perturbation matrix by solving an optimization problem with a soft constraint. Moreover, our method is not restricted to the HiPPO matrix but can be applied to any initializations. **(5)** We provide an ablation study for the size of the perturbation in our models. We also evaluate our S4-PTD and S5-PTD models on LRA tasks, which reveals that the S4-PTD model outperforms the S4D model, while the S5-PTD model is comparable with the S5 model (see section 5.1).

## 2 Preliminaries and notation

Given an LTI system in eq. (1), we say it is asymptotically stable if the eigenvalues $\lambda_j$ of $\mathbf{A}$ are all contained in the left half-plane, i.e., if $\text{Re}(\lambda_j) < 0$ for all $1 \leq j \leq n$. The *transfer function* of the LTI system is defined by

$$G(s) = \mathbf{C}(s\mathbf{I} - \mathbf{A})^{-1}\mathbf{B} + \mathbf{D}, \qquad s \in \mathbb{C} \setminus \Lambda(\mathbf{A}), \tag{2}$$

where $\mathbf{I} \in \mathbb{R}^{n \times n}$ is the identity matrix and $\Lambda(\mathbf{A})$ is the spectrum of $\mathbf{A}$. The transfer function $G$ is a rational function with $n$ poles (counting multiplicities) at the eigenvalues of $\mathbf{A}$. Assume $\mathbf{x}(0) = \mathbf{0}$. Then the transfer function maps the inputs to the outputs of the LTI system in the Laplace domain by multiplication, i.e., $(\mathcal{L}\mathbf{y})(s) = G(s)(\mathcal{L}\mathbf{u})(s)$ for all $s \in \mathbb{C}$, where $\mathcal{L}$ is the Laplace transform operator (see Zhou & Doyle (1998)). Assume the LTI system in eq. (1) is asymptotically stable and the input $\mathbf{u}(t)$ is bounded and integrable (with respect to the Lebesgue measure) as $t$ ranges over $\mathbb{R}$. Then the Laplace transform reduces to the Fourier transform:

$$\hat{\mathbf{y}}(s) = G(is)\hat{\mathbf{u}}(s), \qquad s \in \mathbb{R}, \tag{3}$$

where $\hat{\mathbf{y}}$ and $\hat{\mathbf{u}}$ are the Fourier transforms of $\mathbf{y}$ and $\mathbf{u}$, respectively, and $i$ is the imaginary unit. Let $\mathbf{V} \in \mathbb{C}^{n \times n}$ be an invertible matrix. We can conjugate the system $(\mathbf{A}, \mathbf{B}, \mathbf{C}, \mathbf{D})$ by $\mathbf{V}$, which yields $(\mathbf{V}^{-1}\mathbf{A}\mathbf{V}, \mathbf{V}^{-1}\mathbf{B}, \mathbf{C}\mathbf{V}, \mathbf{D})$. Since the transfer function is conjugation-invariant, the two systems map the same inputs $\mathbf{u}(\cdot)$ to the same outputs $\mathbf{y}(\cdot)$, while the states $\mathbf{x}(\cdot)$ are transformed by $\mathbf{V}$. If $\mathbf{A}$ is a normal matrix, i.e., $\mathbf{A}\mathbf{A}^* = \mathbf{A}^*\mathbf{A}$, then $\mathbf{V}$ is unitary, in which case transforming the states by $\mathbf{V}$ is a well-conditioned problem and can be done without loss of information. Issues arise, however, when $\mathbf{A}$ is non-normal and $\mathbf{V}$ is ill-conditioned.

The state-space models use LTI systems to process time series inputs. Different initializations can be tailored to tasks with different natures, such as the range of dependency (Gu et al., 2023). A particularly successful initialization scheme used in the S4 model is the so-called HiPPO initialization. While there exist several variants of HiPPO, the most popular HiPPO-LegS matrices are defined by

$$(A_H)_{jk} = -\begin{cases} \mathbb{1}_{\{j>k\}}\sqrt{2j-1}\sqrt{2k-1} & , & \text{if } j \neq k, \\ j & , & \text{if } j = k, \end{cases} \qquad (B_H)_{j\ell} = \sqrt{(2j-1)/2}, \tag{4}$$

for all $1 \leq j, k \leq n$ and $1 \leq \ell \leq m$, where $\mathbb{1}_{\{j>k\}}$ is the indicator that equals 1 if $j > k$ and 0 otherwise. Such a system guarantees that the Legendre coefficients of the input history $\mathbf{u} \cdot \mathbb{1}_{[0,t]}$ (with respect to a scaled measure) are stored in the states $\mathbf{x}(t)$ over time (Gu et al., 2020). Since computing with the dense matrix $\mathbf{A}_H$ is practically inefficient, one conjugates the HiPPO system with a matrix $\mathbf{V}_H$ to simplify the structure of $\mathbf{A}_H$. The matrix $\mathbf{A}_H$ in eq. (4) has an ill-conditioned eigenvector matrix (Gu et al., 2022b); consequently, instead of solving the ill-posed problem that diagonalizes $\mathbf{A}_H$, one exploits a diagonal-plus-low-rank (DPLR) structure:

$$\mathbf{A}_H = \mathbf{A}_H^\perp - \frac{1}{2}\mathbf{B}_H\mathbf{B}_H^\top, \qquad (A_H^\perp)_{jk} = -\frac{1}{2}\begin{cases} (-1)^{\mathbb{1}_{\{j<k\}}}\sqrt{2j-1}\sqrt{2k-1} & , & j \neq k, \\ 1 & , & j = k, \end{cases} \tag{5}$$

where $\mathbf{A}_H^\perp$ is a skew-symmetric matrix that can be unitarily diagonalized into $\mathbf{A}_H^\perp = \mathbf{V}_H \mathbf{\Lambda}_H \mathbf{V}_H^{-1}$. The S4 model leverages the HiPPO matrices by initializing

$$\mathbf{A}_{\mathrm{DPLR}} = \mathbf{\Lambda}_H - \frac{1}{2} \mathbf{V}_H^{-1} \mathbf{B}_H \mathbf{B}_H^\top \mathbf{V}_H, \qquad \mathbf{B}_{\mathrm{DPLR}} = \mathbf{V}_H^{-1} \mathbf{B}_H \qquad (6)$$

and $\mathbf{C}_{\mathrm{DPLR}}$ and $\mathbf{D}_{\mathrm{DPLR}}$ randomly. Such an LTI system $\Sigma_{\mathrm{DPLR}} = (\mathbf{A}_{\mathrm{DPLR}}, \mathbf{B}_{\mathrm{DPLR}}, \mathbf{C}_{\mathrm{DPLR}}, \mathbf{D}_{\mathrm{DPLR}})$ is conjugate via $\mathbf{V}_H$ to $(\mathbf{A}_H, \mathbf{B}_H, \mathbf{C}_{\mathrm{DPLR}} \mathbf{V}_H^{-1}, \mathbf{D}_{\mathrm{DPLR}})$. Hence, they share the transfer function and the same mapping from the inputs $\mathbf{u}(\cdot)$ to the outputs $\mathbf{y}(\cdot)$. The S4D model further simplifies the structure by discarding the rank-1 part from $\mathbf{A}_H$ and therefore initializes

$$\mathbf{A}_{\mathrm{Diag}} = \mathbf{\Lambda}_H, \qquad \mathbf{B}_{\mathrm{Diag}} = \frac{1}{2} \mathbf{V}_H^{-1} \mathbf{B}_H, \qquad (7)$$

and $\mathbf{A}_{\mathrm{Diag}}$ is henceforth restricted to be diagonal. While both the S4 and S4D models restrict that $m = p = 1$, i.e., the LTI systems are single-input/single-output (SISO), the S5 model, which also initializes $\mathbf{A}_{\mathrm{Diag}} = \mathbf{\Lambda}_H$ and requires it to be diagonal throughout training, leverages multiple-input/multiple-output (MIMO) systems by allowing $m = p > 1$. We provide more background information on LTI systems and state-space models in sequential modeling in Appendix B.

Throughout this paper, we use $\|\cdot\|$ to denote a vector or matrix 2-norm. Given an invertible square matrix $\mathbf{V}$, we use $\kappa(\mathbf{V}) = \|\mathbf{V}\| \|\mathbf{V}^{-1}\|$ to denote its condition number. Given a number $1 \le p \le \infty$ and a measurable function $f : \mathbb{R} \to \mathbb{C}$, we use $\|f\|_{L^p}$ for the standard $L^p$-norm of $f$ with respect to the Lebesgue measure on $\mathbb{R}$ and $L^p(\mathbb{R}) = \{f : \mathbb{R} \to \mathbb{C} \mid \|f\|_{L^p} < \infty\}$.

## 3 THEORY OF THE DIAGONAL INITIALIZATION OF STATE-SPACE MODELS

The S4 model proposes to initialize the SSM to store the Legendre coefficients of the input signal in the states $\mathbf{x}$ (Gu et al., 2020). This initialization, however, has an ill-conditioned spectrum, preventing a stable diagonalization of the SSM. On the other hand, the S4D model uses a different initialization scheme that has a stable spectrum, allowing for stable diagonalization; however, such initialization lacks an interpretation of the states $\mathbf{x}$. In this section, we conduct a fine-grained analysis of the two initializations, which shows that: (1) for any fixed input signal $\mathbf{u}(\cdot)$ with sufficient smoothness, the outputs of the two systems $\Sigma_{\mathrm{DPLR}}$ and $\Sigma_{\mathrm{Diag}}$ converge to each other with a linear rate (of which the previous analysis is devoid) as $n \to \infty$; and (2) by viewing $\Sigma_{\mathrm{DPLR}}$ and $\Sigma_{\mathrm{Diag}}$ as linear operators that map input signals to the outputs, the operators do not converge in the operator norm topology as $n \to \infty$ (see section 3.1). While the first observation partially justifies the success of the S4D model, the second one allows us to observe that the diagonal initialization is unstable under certain Fourier-mode input perturbations (see section 5.2). In this section, we assume $m = p = 1$, which is consistent with the S4 and S4D models. Still, our theory can be related to the S5 model, as shown in Smith et al. (2023).

Fix an integer $1 \le \ell \le n$. We assume that $\mathbf{C}_{\mathrm{DPLR}} = \mathbf{C}_{\mathrm{Diag}} = \mathbf{e}_\ell^\top \mathbf{V}_H$, where $\mathbf{e}_\ell^\top$ is the $\ell$th standard basis, and $\mathbf{D}_{\mathrm{DPLR}} = \mathbf{D}_{\mathrm{Diag}}$. For a general $\mathbf{C}_{\mathrm{DPLR}} = \mathbf{C}_{\mathrm{Diag}}$, we can decompose it onto the orthonormal basis $\{\mathbf{e}_\ell^\top \mathbf{V}_H \mid 1 \le \ell \le n\}$ and study each component separately using the theory developed in this section. Let $G_{\mathrm{DPLR}}$ and $G_{\mathrm{Diag}}$ be the transfer functions of $\Sigma_{\mathrm{DPLR}}$ and $\Sigma_{\mathrm{Diag}}$, respectively, i.e.,

$$G_{\mathrm{DPLR}}(s) = \mathbf{C}_{\mathrm{DPLR}}(s\mathbf{I} - \mathbf{A}_{\mathrm{DPLR}})^{-1} \mathbf{B}_{\mathrm{DPLR}} + \mathbf{D}_{\mathrm{DPLR}}, \quad G_{\mathrm{Diag}}(s) = \mathbf{C}_{\mathrm{Diag}}(s\mathbf{I} - \mathbf{A}_{\mathrm{Diag}})^{-1} \mathbf{B}_{\mathrm{Diag}} + \mathbf{D}_{\mathrm{Diag}}. \quad (8)$$

Recall that by eq. (3), $|G_{\mathrm{DPLR}}(si) - G_{\mathrm{Diag}}(si)|$ measures the difference between the outputs of the two systems given a frequency-$s$ input. We provide a fine-grained analysis of this difference in the two transfer functions in Lemma 1. The lemma is visualized in Figure 1. We see that as $n$ increases, $G_{\mathrm{Diag}}$ approaches $G_{\mathrm{DPLR}}$ in the low-frequency domain, i.e., when $|s|$ is small. However, $G_{\mathrm{Diag}}$ develops spikes in the high-frequency domain. Moreover, for every $n \ge 1$, zooming into the last spike located at $|s| = \Theta(n^2)$ reveals that it has a constant magnitude (see the subplots on the right in Figure 1). Hence, the convergence of $G_{\mathrm{Diag}}$ to $G_{\mathrm{DPLR}}$ is non-uniform (see Theorem 2). Moreover, the frequency response is unstable at input frequencies $s$ near these spikes, suggesting that the S4D model is not robust to certain input perturbations (see section 5.2).

### 3.1 INPUT-WISE CONVERGENCE AND SYSTEM-WISE DIVERGENCE OF THE DIAGONAL INITIALIZATION

First, we present a result to show that for a fixed input signal $\mathbf{u}(\cdot)$, the outputs of $\Sigma_{\mathrm{DPLR}}$ and $\Sigma_{\mathrm{Diag}}$ converge to each other as $n \to \infty$. Moreover, while the previous result in Gu et al. (2022a) does not

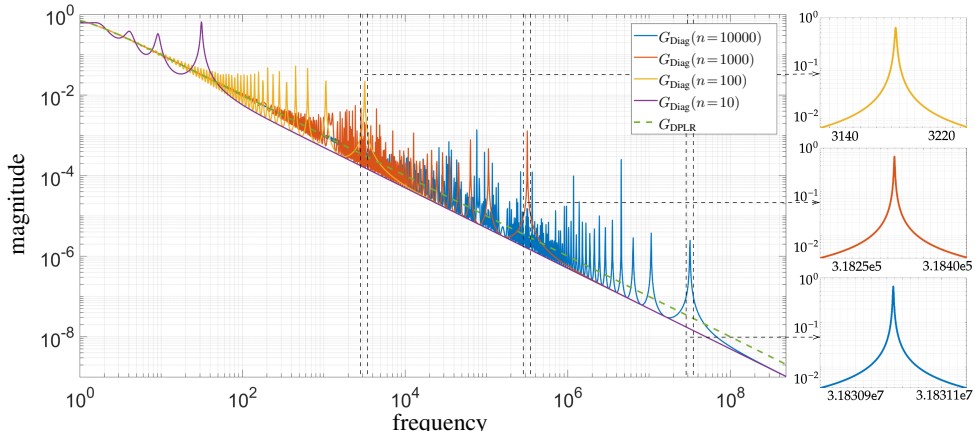

**Figure 1:** The magnitude of transfer function of the S4 model, $|G_{\text{DPLR}}(si)|$, and that of the S4D model, $|G_{\text{Diag}}(si)|$ with $\mathbf{C}_{\text{DPLR}} = \mathbf{C}_{\text{Diag}} = \mathbf{e}_1^\top \mathbf{V}_H$ and the SSM size $n$ set to different values. Note that $G_{\text{DPLR}}$ stays the same regardless of $n$. Due to the limited resolution, the left panel does not correctly reveal the heights of the spikes; however, by zooming into the last spike of $|G_{\text{Diag}}(si)|$, we see that the peak remains $\Theta(1)$ as $n \to \infty$ (see the right panels). The figure shows that $G_{\text{Diag}}$ is oscillatory while $G_{\text{DPLR}}$ is smooth; moreover, $|G_{\text{Diag}}(si)|$ does not converge to $|G_{\text{DPLR}}(si)|$ uniformly.

have a rate of convergence, we show that it is linear. In fact, the rate is sharp (see Appendix F). This partially explains why the S4D model matches the performance of the S4 model in practice.

**Theorem 1.** Let $\mathbf{u}(\cdot) \in L^2(\mathbb{R})$ be an input function with $\|\mathbf{u}\|_{L^2} = 1$. Let $\mathbf{y}_{\text{DPLR}}(\cdot)$ and $\mathbf{y}_{\text{Diag}}(\cdot)$ be the outputs of $\Sigma_{\text{DPLR}}$ and $\Sigma_{\text{Diag}}$ given the input $\mathbf{u}(\cdot)$ and the initial states $\mathbf{x}(0) = \mathbf{0}$, respectively. For some $q > 1/2$, suppose $|\hat{\mathbf{u}}(s)| = \mathcal{O}(|s|^{-q})$ as $|s| \to \infty$. Then, we have $\|\mathbf{y}_{\text{DPLR}} - \mathbf{y}_{\text{Diag}}\|_{L^2} = \mathcal{O}(n^{-1})\sqrt{\ell}$ as $n \to \infty$, where the constant in the $\mathcal{O}$-notation only depends on $q$ and the constant in $\hat{\mathbf{x}}(s) = \mathcal{O}(|s|^{-q})$. The constant does not depend on $q$ if we restrict $q \in [q', \infty)$ for a fixed $q' > 1/2$.

The proof is deferred to Appendix E. Since the Fourier transform interchanges smoothness and decay, what Theorem 1 says is that under a mild assumption that $\mathbf{u}(\cdot)$ is sufficiently smooth, the output of the diagonal system converges linearly to that of the DPLR system as $n \to \infty$. In Section 3.2, we show this smoothness assumption is needed. We know the two systems converge input-wise; it is natural to ask if the convergence is uniform across all input signals:

**Theorem 2.** The function $G_{\text{DPLR}}(s) - G_{\text{Diag}}(s)$ does not converge to zero uniformly on the imaginary axis as $n \to \infty$. In particular, for every $n \geq 1$, there exists an input signal $\mathbf{u}_n(\cdot) \in L^1(\mathbb{R}) \cap L^2(\mathbb{R})$ such that if we let $\mathbf{y}_{n,\text{DPLR}}$ and $\mathbf{y}_{n,\text{Diag}}$ be the outputs of $\Sigma_{\text{DPLR}}$ and $\Sigma_{\text{Diag}}$ of degree $n$, respectively, then we have $\|\mathbf{y}_{n,\text{DPLR}} - \mathbf{y}_{n,\text{Diag}}\|_{L^2}$ does not converge to 0 as $n \to \infty$.

Hence, the answer to our question is negative: combined with Theorem 1, Theorem 2 says that while a sufficiently large S4D model mimics its S4 alternative on a fixed smooth input, when we predetermine a size $n$, they inevitably disagree, by a large amount, on some inputs. Moreover, in Theorem 2, the construction of $\mathbf{u}_n(\cdot)$ can be made explicit (see section 5.2).

## 3.2 SOME NUMERICAL EXAMPLES

In this section, we provide some numerical examples corroborating Theorem 1. We defer the implication of Theorem 2 to later sections (see section 4 and section 5.2). Theorem 1 tells us that if we fix a smooth input signal $\mathbf{u}(t)$, then the outputs $\mathbf{y}_{n,\text{DPLR}}$ and $\mathbf{y}_{n,\text{Diag}}$ eventually converge to each other at a linear rate as $n \to \infty$. In this experiment, we fix two input functions (or more precisely, distributions)

$$\mathbf{u}_e(t) = e^{-t}H(t), \qquad \mathbf{u}_d = \delta_0,$$

where $H = \mathbb{1}_{[0,\infty)}$ is the Heaviside function and $\delta_0$ is the Dirac delta function at 0. While $\mathbf{u}_e(t)$ is a very smooth function — in particular, we have $|\hat{\mathbf{u}}_e(s)| = \mathcal{O}(|s|^{-1})$ — the Dirac delta $\mathbf{u}_d$ is very non-smooth with a Fourier transform that is constantly one. We simulate both systems $\Sigma_{\text{DPLR}}$

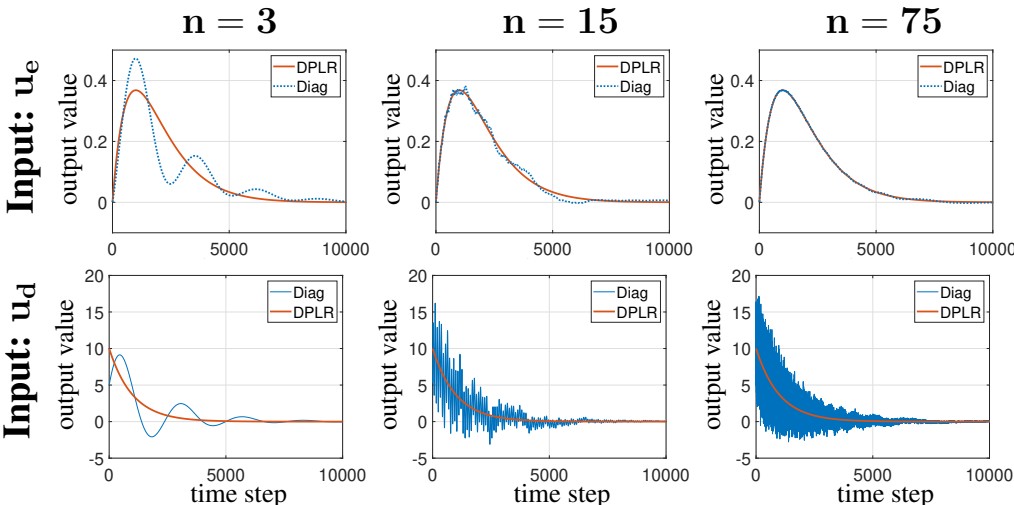

**Figure 2:** Simulated outputs of the DPLR and diagonal systems with the input functions $\mathbf{u}_e$ and $\mathbf{u}_d$ and varying state-space dimension $n$. We see that for a smooth input function $\mathbf{u}_e$, the outputs of both systems converge rapidly as $n$ increases, whereas the convergence does not happen for a non-smooth input function $\mathbf{u}_d$.

and $\Sigma_{\text{Diag}}$ on both $\mathbf{u}_e(t)$ and $\mathbf{u}_d(t)$. More details of the simulation can be found in Appendix F. From Figure 2, we observe that given a smooth input function $\mathbf{u}_e$, the output $\mathbf{y}_{n,\text{Diag}}$ converges to $\mathbf{y}_{n,\text{DPLR}}$ rapidly, but the same does not hold for a non-smooth input function $\mathbf{u}_d$. Hence, the smoothness assumption in Theorem 1 is essential. In Figure 8 in Appendix F, we also compute the $L^2$-norm of $\mathbf{y}_{n,\text{DPLR}} - \mathbf{y}_{n,\text{Diag}}$ and verify that the convergence is linear when the input is smooth enough. We remark that a similar study of $\mathbf{u}_d$ can be found in Gu et al. (2022a), where the results appear qualitatively different from those presented in Figure 2. This does not mean either work is wrong: the key distinction is that the discretization step size of the LTI systems (see Appendix B) is fixed in Gu et al. (2022a) *a priori*, introducing aliasing errors and hiding the high frequencies (Trefethen, 2019, Ch. 4.). Consequently, when $n$ is large, the difference between $G_{\text{DPLR}}$ and $G_{\text{Diag}}$ in the high-frequency domain is overlooked. In comparison, in this paper, our theory considers the continuous-time LTI systems, which take every mode into account.

## 4 PERTURBING THE HIPPO INITIALIZATION: A NEW WAY OF DIAGONALIZING THE STATE-SPACE MODEL

In section 3, we saw the instability of the S4D transfer function at certain Fourier modes. Nevertheless, the diagonal structure of $\mathbf{A}$ is preferred over the DPLR one due to its training and inference efficiency and its adaptivity to the MIMO model (i.e., the S5 model) (Smith et al., 2023). To avoid instability in a diagonal model, we want to leverage the HiPPO initialization in eq. (4) instead of the one in eq. (7) that discards the rank-1 part. One obvious solution is to diagonalize the HiPPO matrix $\mathbf{A}_H = \mathbf{V}_H \mathbf{\Lambda}_H \mathbf{V}_H^{-1}$ and conjugate $(\mathbf{A}_H, \mathbf{B}_H, \mathbf{C}, \mathbf{D})$ using $\mathbf{V}_H$. However, as shown in Gu et al. (2022a), the eigenvector matrix $\mathbf{V}_H$ is exponentially ill-conditioned with respect to $n$, making the spectral information meaningless. While the exact eigenvalues and eigenvectors of $\mathbf{A}_H$ are very ill-conditioned, since we only care about the backward error of diagonalization, we propose the following initialization scheme. let $\mathbf{E} \in \mathbb{C}^{n \times n}$ be a perturbation matrix. We diagonalize the perturbed HiPPO matrix as

$$\tilde{\mathbf{A}}_H = \mathbf{A}_H + \mathbf{E} = \tilde{\mathbf{V}}_H \tilde{\mathbf{\Lambda}}_H \tilde{\mathbf{V}}_H^{-1}. \tag{9}$$

We then initialize the systems using $\Sigma_{\text{Pert}} = (\mathbf{A}_{\text{Pert}}, \mathbf{B}_{\text{Pert}}, \mathbf{C}_{\text{Pert}}, \mathbf{D}_{\text{Pert}}) = (\tilde{\mathbf{\Lambda}}_H, \tilde{\mathbf{V}}_H^{-1} \mathbf{B}_H, \mathbf{C}, \mathbf{D})$, where $\mathbf{C}$ and $\mathbf{D}$ are random matrices. Therefore, we approximately diagonalize the HiPPO initialization in the sense that although the diagonal entries in $\tilde{\mathbf{\Lambda}}$ do not approximate the eigenvalues of $\mathbf{A}_H$, the transfer function of $\Sigma_{\text{Pert}}$ is an approximation of that of $\Sigma_{\text{DPLR}}$ (see Theorem 3). We call our model S4-PTD or S5-PTD, depending on whether the model architecture is adapted from the S4D or the S5 model, where "PTD" stands for "perturb-then-diagonalize." Since our models are only different from the S4D and the S5 models in initialization, we refer interested readers to Gu et al. (2022a)

and Smith et al. (2023) for a discussion of computation details and time/space complexity. Our proposed perturb-then-diagonalize method is not restricted to the HiPPO-LegS matrices in eq. (4). This endows our method with adaptivity to any (dense) initialization scheme. This adaptivity was absent from the previous line of work on SSMs. Consider the process of diagonalizing the matrix $\mathbf{A}_H = \mathbf{V}_H \mathbf{\Lambda}_H \mathbf{V}_H^{-1}$ that is solved by an inexact algorithm. In a numerical analyst's language, the forward error is the error made in computing the eigenvalues $\mathbf{\Lambda}_H$ and eigenvectors $\mathbf{V}_H$, whereas the backward error asks how close a problem that we have solved exactly (i.e., $\mathbf{A}_H + \mathbf{E}$) is to the actual problem that we want to solve (i.e., $\mathbf{A}_H$). As we will see in Theorem 3, it is the backward error $\|\mathbf{E}\|$ (but not the forward error) that matters in our initialization because it is the matrix $\mathbf{A}_H$ (but not the specific forms of $\mathbf{V}_H$ or $\mathbf{\Lambda}_H$) that is important in the transfer function.

Centered around the perturbed initialization scheme eq. (9) are two important questions: (1) What is the difference between the perturbed initialization $(\mathbf{A}_{\text{Pert}}, \mathbf{B}_{\text{Pert}}, \mathbf{C}_{\text{Pert}}, \mathbf{D}_{\text{Pert}})$ and the HiPPO initialization $(\mathbf{A}_{\text{DPLR}}, \mathbf{B}_{\text{DPLR}}, \mathbf{C}_{\text{DPLR}}, \mathbf{D}_{\text{DPLR}})$? (2) What is the condition number of $\tilde{\mathbf{V}}_H$? The first question is important because it controls the deviation of our perturbed initialization from the successful and robust DPLR initialization. The second question is important because it shadows the numerical robustness of conjugating the LTI system by $\tilde{\mathbf{V}}_H$. Moreover, since the state vector $\mathbf{x}(t)$ is transformed by $\tilde{\mathbf{V}}_H$ via conjugation (see section 2), a small condition number of $\tilde{\mathbf{V}}_H$ shows that its singular values are more evenly distributed. Hence, the transformation $\tilde{\mathbf{V}}_H$ does not significantly magnify or compress $\mathbf{x}(t)$ onto some particular modes. To study the first question, we define the transfer function of the perturbed system to be

$$G_{\text{Pert}}(s) = \mathbf{C}_{\text{Pert}}(s\mathbf{I} - \mathbf{A}_{\text{Pert}})^{-1}\mathbf{B}_{\text{Pert}} + \mathbf{D}_{\text{Pert}}.$$

We control the size of the transfer function perturbation by proving the following theorem.

**Theorem 3.** Assume $\mathbf{C}_{\text{Pert}}\tilde{\mathbf{V}}_H^{-1} = \mathbf{C}_{\text{DPLR}}\mathbf{V}_H^{-1}$ and $\mathbf{D}_{\text{Pert}} = \mathbf{D}_{\text{DPLR}}$. Suppose $\|\mathbf{E}\| \le \epsilon$ and we normalize the matrices so that $\|\tilde{\mathbf{V}}_H \mathbf{B}_{\text{Pert}}\| = \|\mathbf{V}_H \mathbf{B}_{\text{DPLR}}\| = \|\mathbf{C}_{\text{Pert}}\tilde{\mathbf{V}}_H^{-1}\| = \|\mathbf{C}_{\text{DPLR}}\mathbf{V}_H^{-1}\| = 1$. For any $s$ on the imaginary axis, we have

$$|G_{\text{Pert}}(s) - G_{\text{DPLR}}(s)| = (2\ln(n) + 4)\epsilon + \mathcal{O}(\sqrt{\log(n)}\epsilon^2).$$

While our perturb-then-diagonalize method works for a general initialization and a bound on the transfer function error can always be established, the proof of Theorem 3 leverages the structure of HiPPO matrices to improve this bound. The error in Theorem 3 is the uniform error on the imaginary axis. Using Hölder's inequality, for any bounded and integrable input function $\mathbf{u}(\cdot)$, if $\mathbf{y}_{\text{Pert}}$ and $\mathbf{y}_{\text{DPLR}}$ are the outputs of $\Sigma_{\text{Pert}}$ and $\Sigma_{\text{DPLR}}$, respectively, then we have

$$\|\mathbf{y}_{\text{Pert}} - \mathbf{y}_{\text{DPLR}}\|_{L^2} = \|\hat{\mathbf{y}}_{\text{Pert}} - \hat{\mathbf{y}}_{\text{DPLR}}\|_{L^2} = \|\hat{\mathbf{x}}(s)(G_{\text{Pert}}(is) - G_{\text{DPLR}}(is))\|_{L^2}$$
$$\le \|\hat{\mathbf{x}}(s)\|_{L^2}\|G_{\text{Pert}}(is) - G_{\text{DPLR}}(is)\|_{L^\infty} \le \|\mathbf{x}\|_{L^2}\big((2\ln(n)+4)\epsilon + \mathcal{O}(\sqrt{\log(n)}\epsilon^2)\big),$$

(10)

where the first and the last steps follow from Parseval's identity. Hence, Theorem 3 gives us an upper bound on the distance between $\Sigma_{\text{Pert}}$ and $\Sigma_{\text{DPLR}}$ in the operator norm topology. The theorem states that the error made by the perturbation is linear in the size of the perturbation. Moreover, the error depends only logarithmically on the dimension $n$ of the state space.

Next, we consider the conditioning of $\tilde{\mathbf{V}}_H$, which affects the accuracy of computing $\tilde{\mathbf{V}}_H^{-1}\mathbf{B}_{\text{Pert}}$ and the scaling ratio of the states in $\mathbf{x}(\cdot)$ (see Appendix B). The following theorem provides a deterministic estimate of the eigenvector condition number for the "best perturbation scheme."

**Theorem 4** ((Banks et al., 2021, Thm. 1.1.)). Given any $\mathbf{A} \in \mathbb{C}^{n \times n}$ and $\epsilon \in (0, 1)$, there exists a matrix $\mathbf{E} \in \mathbb{C}^{n \times n}$ with $\|\mathbf{E}\| \le \epsilon$ and an eigenvector matrix $\tilde{\mathbf{V}}$ of $\mathbf{A} + \mathbf{E}$ such that

$$\kappa(\tilde{\mathbf{V}}) \le 4n^{3/2}\big(1 + \epsilon^{-1}\|\mathbf{A}\|\big).$$

Theorem 4 shows the promise of finding a good perturbation matrix to reduce the eigenvector condition number. We remark that while Theorem 4 studies the best-case scenario, Banks et al. (2021) also contains a probabilistic statement about Gaussian perturbations (see Appendix H). In this paper, we propose to compute $\mathbf{E}$ by solving the following optimization problem with a soft constraint:

$$\text{minimize } \Phi(\mathbf{E}) = \kappa(\tilde{\mathbf{V}}_H) + \gamma\|\mathbf{E}\| \qquad \text{s.t.} \qquad \mathbf{A}_H + \mathbf{E} = \tilde{\mathbf{V}}_H \tilde{\mathbf{\Lambda}} \tilde{\mathbf{V}}_H^{-1}, \quad \tilde{\mathbf{\Lambda}} \text{ diagonal,} \qquad (11)$$

where $\gamma > 0$ is a hyperparameter that controls the trade-off between $\kappa(\tilde{\mathbf{V}}_H)$ and $\|\mathbf{E}\|$. We implement a solver to this optimization problem using gradient descent. As $\gamma$ increases, it is harder to recover the original states $\mathbf{x}(\cdot)$ from the transformed states $\tilde{\mathbf{V}}_H\mathbf{x}(\cdot)$ because $\kappa(\tilde{\mathbf{V}}_H)$ increases, but $\|\mathbf{E}\|$ decreases, resulting in a more robust SSM that is closer to the flawless HiPPO initialization.

| Model | ListOps | Text | Retrieval | Image | Pathfinder | Path-X | Avg. |
|---|---|---|---|---|---|---|---|
| Transformer | 36.37 | 64.27 | 57.56 | 42.44 | 71.40 | ✗ | 53.66 |
| Luna-256 | 37.25 | 64.57 | 79.29 | 47.38 | 77.72 | ✗ | 59.37 |
| H-Trans.-1D | 49.53 | 78.69 | 63.99 | 46.05 | 68.78 | ✗ | 61.41 |
| CCNN | 43.60 | 84.08 | ✗ | 88.90 | 91.51 | ✗ | 68.02 |
| S4 | 59.60 | 86.82 | 90.90 | 88.65 | 94.20 | 96.35 | 86.09 |
| Liquid-S4 | **62.75** | 89.02 | 91.20 | **89.50** | 94.80 | 96.66 | 87.32 |
| S4D | 60.47 | 86.18 | 89.46 | 88.19 | 93.06 | 91.95 | 84.89 |
| S4-PTD (ours) | 60.65 | 88.32 | 91.07 | 88.27 | 94.79 | 96.39 | 86.58 |
| S5 | 62.15 | 89.31 | 91.40 | 88.00 | 95.33 | **98.58** | 87.46 |
| S5-PTD (ours) | **62.75** | **89.41** | **91.51** | 87.92 | **95.54** | 98.52 | **87.61** |

**Table 1:** Test accuracies on LRA, where ✗ means the model isn't outperforming random guessing. We use the boldface number to indicate the highest test accuracy among all models for each task. We use the underlined number to indicate the highest test accuracy within the comparable group.

## 5 EMPIRICAL EVALUATION AND DISCUSSION

In this section, we present empirical evaluations of our proposed S4-PTD and S5-PTD models. In section 5.1 we compare the performance of our full model with the existing ones in the Long Range Arena (LRA). In section 5.2, we perform a sensitivity analysis using the CIFAR-10 dataset to provide real-world evidence that our perturbed initialization scheme is more robust than the one in the S4D/S5 model. Finally, in section 5.3, we study the relationship between the size of the perturbation matrix $\mathbf{E}$ and the performance of our models.

### 5.1 PERFORMANCE IN THE LONG-RANGE ARENA

The LRA benchmark comprises six tasks with sequential data (Tay et al., 2021). This collection, with its sequence lengths ranging from 1024 to 16000, is designed to measure the model's capability of processing the long-range inputs. We train an S4-PTD model and an S5-PTD model to learn these tasks, respectively. We adopt the same SSM architectures, and thus the same number of parameters, from the original S4D (Gu et al., 2022a) and S5 papers (Smith et al., 2023). Results are reported in Table 1, along with the accuracies of other sequential models, including the Liquid-S4 model which is built upon S4 (Hasani et al., 2023). We report details of hyperparameters in Appendix J. While the perturbation matrix $\mathbf{E}$ is also tunable, we restrict its size to be less than 10% of that of the HiPPO matrix $\mathbf{A}_H$, promoting the worst-case robustness of our model (see section 5.2). We note that the S4-PTD model outperforms the S4D model[3] (and even the S4 model with the DPLR structure for most tasks), while the S5-PTD model matches the performance of the S5 model.

### 5.2 ROBUSTNESS OF OUR PERTURBED MODEL OVER THE DIAGONAL MODEL

Our discussion in section 3 suggests that the S4D initialization is not as stable as the S4 initialization (see Figure 1). Here, we demonstrate its practical implication regarding the robustness of the model. We train an S4D model and an S4-PTD model (with $\|\mathbf{E}\|/\|\mathbf{A}_H\| \approx 10^{-1}$) to learn the sCIFAR task, where the images in the CIFAR-10 dataset (Krizhevsky et al., 2009) are flattened into sequences of pixels. We test the two models against two different test sets: one is taken from the original CIFAR-10 dataset while the other one is contaminated by 10% of sinusoidal noises whose frequencies are located near the spikes of $|G_{\text{Diag}}|$. We plot the training and test accuracies of the two models in Figure 3a and b. Whereas the two models both achieve high accuracies on the uncontaminated test set, the S4D model does not generalize to the noisy dataset as the S4-PTD model does. That is, the S4D model is not robust to these noises. In comparison, since the S4-PTD initialization is uniformly close to the S4 initialization (see Theorem 3) when $\|\mathbf{E}\|$ is small, the S4-PTD model is robust to noises with any mode. We also perturb the test dataset using noises at different frequencies. In Figure 4, we verify that it is indeed the spikes in $|G_{\text{Diag}}|$ that makes the S4D initialization not robust. We make two remarks. First, the noises in Figure 3a are the "worst-case" noises and intentionally made to fail the S4D model; in practice, the distribution of sensitive modes of S4D in the frequency domain

---

[3]In Orvieto et al. (2023), the S4D model was carefully tuned to have higher accuracies. Since the model architecture does not align with those used in this work, we only report the result from the original S4D paper.

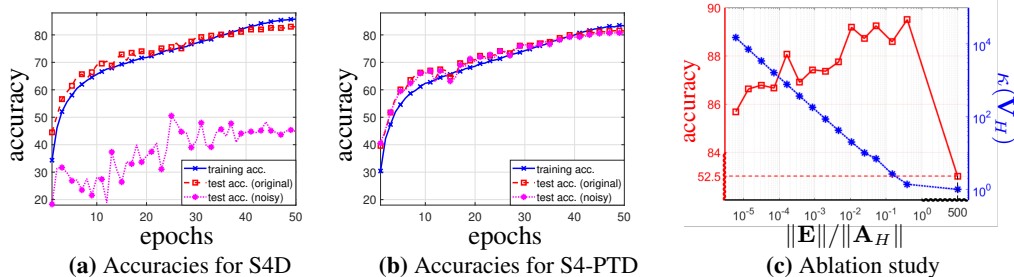

**(a)** Accuracies for S4D     **(b)** Accuracies for S4-PTD     **(c)** Ablation study

**Figure 3:** (a) and (b): the training and test accuracies of the S4D model and the S4-PTD model on contaminated and uncontaminated CIFAR-10 dataset (see section 5.2). (c): The effect of the perturbation size on the accuracy (shown in red) of the S4-PTD model and the eigenvector condition number (shown in blue) of the perturbed HiPPO matrix (see section 5.3).

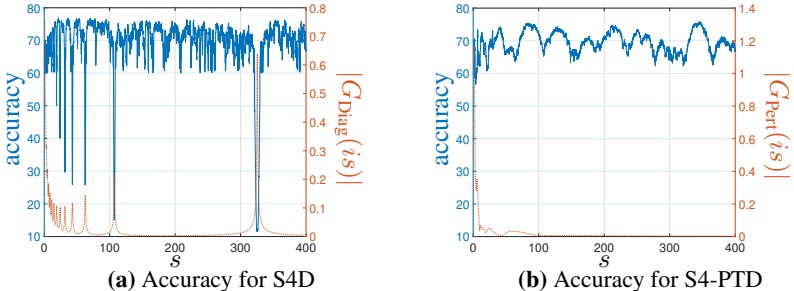

**(a)** Accuracy for S4D        **(b)** Accuracy for S4-PTD

**Figure 4:** A Fourier-mode perturbation of frequency $s$ is added to the test dataset. The blue curve, plotted to the left vertical axis, shows the test accuracy of the model under perturbations of different frequencies $s$. The orange curve, plotted to the right vertical axis, shows the magnitude of the transfer function of the LTI system associated with the initialization.

gets sparser as $n$ increases (see Figure 1), which improves its "average-case" robustness. Also, to enable easy detection of frequencies at which the S4D is unstable, in this experiment, we fix the state matrix $\mathbf{A}$. However, we empirically observed that training the state matrix $\mathbf{A}$ does not resolve the robustness issue. We provide more details about these two remarks in Appendix K.2.

## 5.3 ABLATION STUDY OF OUR MODEL

As mentioned in section 4, the size of the perturbation plays a key role in the performance of our S4-PTD and S5-PTD models. When $\mathbf{E} = 0$, the eigenvector condition number of $\mathbf{A}_H$ is exponential in $n$, making it numerically impossible to diagonalize when $n$ is moderately large. On the other hand, when $\mathbf{E}$ overshadows $\mathbf{A}_H$, the initialization scheme becomes a random one, often leading to poor performance (Gu et al., 2021). In this section, we train an S4-PTD model to learn the sequential CIFAR (sCIFAR) task. We control the size of the perturbation $\|\mathbf{E}\|$ by changing the hyperparameter $\gamma$ in the optimization problem eq. (11). For each perturbation matrix $\mathbf{E}$, we then initialize our S4-PTD model by diagonalizing $\mathbf{A}_H + \mathbf{E}$. In Figure 3c, we plot (in red) the test accuracies with respect to different perturbation sizes. We see that our S4-PTD model achieves its best performance when the ratio between the perturbation size and the size of the HiPPO matrix is between $10^{-2}$ and 1, while the accuracy drops when this ratio gets too small or too large. This aligns with our expectations. In addition, the (blue) curve of the eigenvector condition number admits a straight-line pattern with a slope of roughly $-1$, corroborating the factor $\epsilon^{-1}$ in Theorem 4.

## 6 CONCLUSION

In this paper, we propose a perturb-then-diagonalize (PTD) methodology that can be used to diagonalize the non-normal HiPPO matrices. Motivated by our theoretical study, we apply the PTD method to robustify the diagonal initialization used in the S4D and S5 models. While our theory focuses on initialization, some empirical evaluations suggest that the PTD method also robustifies the trained diagonal models, which is an interesting future research avenue.

ACKNOWLEDGMENTS

This work was supported by the U.S. Department of Energy, Office of Science, Office of Advanced Scientific Computing Research, Scientific Discovery through Advanced Computing (SciDAC) program, under Contract Number DE-AC02-05CH11231 at Lawrence Berkeley National Laboratory. It used the Lawrencium computational cluster provided by the IT Division at the Lawrence Berkeley National Laboratory (Supported by the Director, Office of Science, Office of Basic Energy Sciences, of the U.S. Department of Energy) and resources of the National Energy Research Scientific Computing Center (NERSC, using award ASCR-ERCAP0023337), a U.S. Department of Energy Office of Science User Facility located at Lawrence Berkeley National Laboratory, both operated under Contract No. DE-AC02-05CH11231. NBE would also like to acknowledge NSF, under Grant No. 2319621, for providing partial support of this work. Our conclusions do not necessarily reflect the position or the policy of our sponsors, and no official endorsement should be inferred.

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

APPENDIX

The Appendix is organized as follows. In Appendix A, we survey the background of ill-posed problems, including conditioning, stability, and backward and forward errors. In Appendix B, we provide more background information on LTI systems, including the derivation of the transfer function, system diagonalization, and system discretization, which provides insights into our theory and models. In Appendix C, we prove a result on the difference between the two transfer functions associated with the S4 initialization and the S4D initialization, respectively. Using this result, we prove Theorem 2 and 1 on the uniform divergence and the input-wise convergence in Appendix D and E, respectively. We then present in Appendix F the details of some numerical experiments corroborating these two theorems and in Appendix G an experiment on a synthetic example that shows the S4D initialization is not as robust as the S4/S4-PTD initialization. In Appendix H, we prove the results in section 4 that are related to perturbing the HiPPO matrix and introduce more background of perturbed diagonalization, which are then verified in Appendix I by a numerical experiment. Finally, in Appendix J we give the details of our experiments in section 5 and in appendix K some supplementary results on the robustness of the S4D/S4-PTD model during training.

## A   MORE BACKGROUND INFORMATION OF ILL-POSED PROBLEMS

To make the phrase "ill-posed" precise, we need to introduce the idea of condition numbers. The conditioning is a property of a problem and it does not depend on the algorithm that we use. Abstractly, we let the problem space $\mathcal{X}$ and the solution space $\mathcal{Y}$ be two normed vector spaces with the norms $\|\cdot\|_{\mathcal{X}}$ and $\|\cdot\|_{\mathcal{Y}}$, respectively. Each element in $x \in \mathcal{X}$ is considered as an instance of the problem and its solution in the solution space $\mathcal{Y}$ is defined by a map $f : \mathcal{X} \to \mathcal{Y}$. For example, if we want to solve a system $\mathbf{Ax} = \mathbf{b}_0$ with different matrices $\mathbf{A}$ and a fixed vector $\mathbf{b}_0$, then we can make $\mathcal{X}$ the space of $n$-by-$n$ matrices and $\mathcal{Y}$ the space of vectors of length $n$. In that case, we have $f(\mathbf{A}) = \mathbf{A}^{-1}\mathbf{b}_0$.[4] Likewise, consider the problem of finding eigenvalues. We can make $\mathcal{X}$ and $\mathcal{Y}$ both equal to the space of $n$-by-$n$ matrices and $f(\mathbf{A}) = \mathbf{\Lambda}$, the eigenvalue matrix of $\mathbf{A}$. Now, given an instance $x \in \mathcal{X}$, we define the (absolute) condition number of problem $f$ at $x$ to be

$$\kappa(x; f) = \lim_{\epsilon \to 0} \sup_{\|\delta x\|_{\mathcal{X}} \leq \epsilon} \frac{\|f(x) - f(x + \delta x)\|_{\mathcal{Y}}}{\|\delta x\|_{\mathcal{X}}}.$$

Intuitively, a large condition number means that if we perturb the problem by a little bit, then the solution may become drastically different. Hence, in general, we do not expect that a solution of an ill-conditioned problem can be found accurately using floating-point arithmetic because a small rounding error has a large effect on the computed solution.

Unlike the conditioning of a problem $x$, stability is a property of an algorithm. Let $\tilde{f} : \mathcal{X} \to \mathcal{Y}$ be an algorithm that solves $f$. We are particularly interested in the "backward stability" of $\tilde{f}$. That is, if for any $x \in \mathcal{X}$, there exists an element $\tilde{x} \in \mathcal{X}$ so that $\tilde{f}(x) = f(\tilde{x})$ and

$$E_b(x) = \frac{\|x - \tilde{x}\|_{\mathcal{X}}}{\|x\|_{\mathcal{X}}}$$

is small, then we say that $\tilde{f}$ is backward stable. Intuitively, this is saying that our algorithm is computing the solution to a nearby problem $\tilde{x}$. Note that this is different from saying that

$$E_f(x) = \frac{\|f(x) - \tilde{f}(x)\|_{\mathcal{Y}}}{\|f(x)\|_{\mathcal{Y}}}$$

is small. This error measures how accurately we solved our problem. The error $E_f$ is called a forward error, while $E_b$ is called a backward error. We can control the forward error using the backward error, and this bound is established through the condition number of the problem. (Intuitively, this says that if a problem is well-conditioned, then a small perturbation to the problem does not change the solution by too much. Hence, a small backward error leads to a small forward error.) The advantage of studying backward stability is two-fold. First, the backward error is decoupled from

---

[4]Of course, this function is not defined at singular matrices, but since they form a Lebesgue null set, $f$ is still defined almost everywhere.

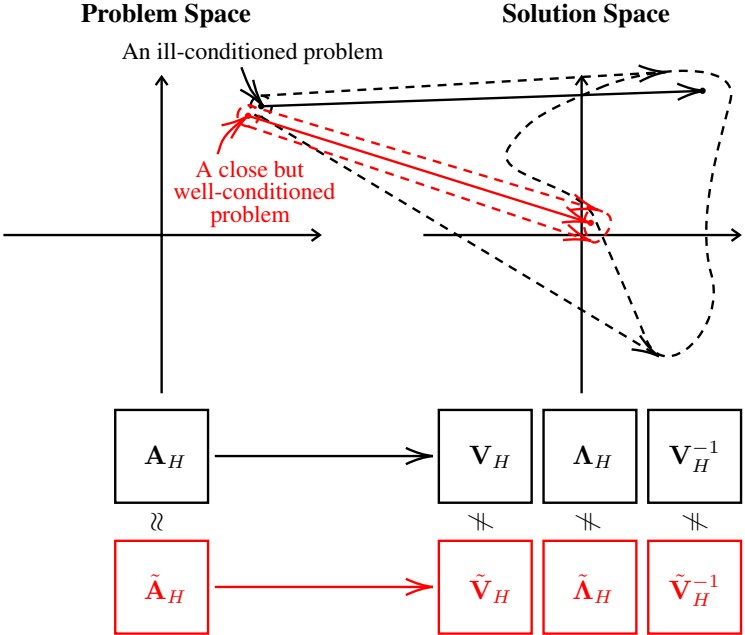

**Figure 5:** An illustration of the perturbation of an ill-posed problem.

the conditioning of the problem. Hence, backward stable algorithms are much more common than forward stable algorithms, because in many cases, the ill-conditioned problems essentially prevent an algorithm from being forward stable. On the other hand, if an algorithm is backward stable, then we know that its forward error must be small on well-conditioned problems.

In our paper, we consider the case where we are forced to solve an ill-conditioned problem $x$. We propose to use a backward stable algorithm $\tilde{f}$ to solve it. Since the problem is ill-conditioned, we do not have that $f(x)$ is close to $\tilde{f}(x)$, i.e., we cannot find the solutions accurately. However, we know that $\tilde{f}(x)$ is the solution to $\tilde{x}$, where $x \approx \tilde{x}$. In many machine learning applications, this is enough to guarantee an acceptable solution. (See Figure 5.)

## B MORE BACKGROUND INFORMATION OF STATE-SPACE MODELS

Recall that an LTI system $\Sigma = (\mathbf{A}, \mathbf{B}, \mathbf{C}, \mathbf{D})$ is given by

$$\begin{aligned} \mathbf{x}'(t) &= \mathbf{A}\mathbf{x}(t) + \mathbf{B}\mathbf{u}(t), \\ \mathbf{y}(t) &= \mathbf{C}\mathbf{x}(t) + \mathbf{D}\mathbf{u}(t). \end{aligned} \tag{12}$$

Assume the initial condition is given by $\mathbf{x}(0) = \mathbf{0}$ and the input function $\mathbf{u}(\cdot)$ is bounded and integrable. Suppose the system is asymptotically stable, i.e., $\Lambda(\mathbf{A})$ is contained in the left half-plane. Then, we have $\mathbf{x}(\cdot)$ and $\mathbf{y}(\cdot)$ are also bounded and integrable. By taking the Fourier transform of the LTI system, we have

$$\begin{aligned} si\hat{\mathbf{x}}(s) &= \mathbf{A}\hat{\mathbf{x}}(s) + \mathbf{B}\hat{\mathbf{u}}(s), \\ \hat{\mathbf{y}}(s) &= \mathbf{C}\hat{\mathbf{x}}(s) + \mathbf{D}\hat{\mathbf{u}}(s). \end{aligned} \tag{13}$$

Rearranging the first equation gives us

$$\hat{\mathbf{x}}(s) = (si\mathbf{I} - \mathbf{A})^{-1}\mathbf{B}\hat{\mathbf{u}}(s), \qquad s \in \mathbb{R}.$$

Plugging it into the second equation of eq. (13), we can derive the transfer function on the imaginary axis:

$$\hat{\mathbf{y}}(s) = \underbrace{\left[\mathbf{C}(si\mathbf{I} - \mathbf{A})^{-1}\mathbf{B} + \mathbf{D}\right]}_{G(si)}\hat{\mathbf{u}}(s), \qquad s \in \mathbb{R}.$$

Let $\mathbf{V} \in \mathbb{C}^{n \times n}$ be an invertible matrix. Consider the system $\tilde{\Sigma} = (\mathbf{V}^{-1}\mathbf{A}\mathbf{V}, \mathbf{V}^{-1}\mathbf{B}, \mathbf{C}\mathbf{V}, \mathbf{D})$:

$$\begin{aligned} \mathbf{x}'(t) &= \mathbf{V}^{-1}\mathbf{A}\mathbf{V}\mathbf{x}(t) + \mathbf{V}^{-1}\mathbf{B}\mathbf{u}(t), \\ \mathbf{y}(t) &= \mathbf{C}\mathbf{V}\mathbf{x}(t) + \mathbf{D}\mathbf{u}(t). \end{aligned} \tag{14}$$

By multiplying the first equation by $\mathbf{V}$, we have

$$\mathbf{V}\mathbf{x}(t) = \mathbf{A}\mathbf{V}\mathbf{x}(t) + \mathbf{B}\mathbf{u}(t),$$

and defining the new state variable $\boldsymbol{\xi}(t) = \mathbf{V}\mathbf{x}(t)$, we can write $\tilde{\Sigma}$ into

$$\boldsymbol{\xi}(t) = \mathbf{A}\boldsymbol{\xi}(t) + \mathbf{B}\mathbf{u}(t),$$
$$\mathbf{y}(t) = \mathbf{C}\boldsymbol{\xi}(t) + \mathbf{D}\mathbf{u}(t).$$

Hence eq. (12) and (14) are equivalent with their states connected via $\mathbf{V}$. We also can verify this by computing the transfer function of $\tilde{\Sigma}$:

$$\tilde{G}(s) := \mathbf{C}\mathbf{V}(s\mathbf{I} - \mathbf{V}^{-1}\mathbf{A}\mathbf{V})^{-1}\mathbf{V}^{-1}\mathbf{B} + \mathbf{D} = \mathbf{C}(s\mathbf{I} - \mathbf{A})^{-1}\mathbf{B} + \mathbf{D} = G(s).$$

The LTI system $\Sigma$ is continuous-time. In order to apply it to sequential input, we need to discretize the system. Given a step size $\Delta t$, there are two common ways of discretizing the system:

$$\text{Bilinear} : \overline{\mathbf{A}} = \left(\mathbf{I} - \frac{\Delta t}{2}\mathbf{A}\right)^{-1}\left(\mathbf{I} + \frac{\Delta t}{2}\mathbf{A}\right), \quad \overline{\mathbf{B}} = \Delta t\left(\mathbf{I} - \frac{\Delta t}{2}\mathbf{A}\right)^{-1}\mathbf{B}, \quad (\overline{\mathbf{C}}, \overline{\mathbf{D}}) = (\mathbf{C}, \mathbf{D}),$$

$$\text{ZOH} : \overline{\mathbf{A}} = \exp(\Delta t\mathbf{A}), \quad \overline{\mathbf{B}} = \mathbf{A}^{-1}(\exp(\Delta t\mathbf{A}) - \mathbf{I})\mathbf{B}, \quad (\overline{\mathbf{C}}, \overline{\mathbf{D}}) = (\mathbf{C}, \mathbf{D}).$$

Then, the discrete system

$$\begin{aligned}
\mathbf{x}_t &= \overline{\mathbf{A}}\mathbf{x}_{t-1} + \overline{\mathbf{B}}\mathbf{u}_{t-1}, \\
\mathbf{y}_t &= \overline{\mathbf{C}}\mathbf{x}_t + \overline{\mathbf{D}}\mathbf{u}_t
\end{aligned} \tag{15}$$

takes the discrete sequential input $(\mathbf{u}_0, \mathbf{u}_1, \ldots)$. The discrete system eq. (15) mimics the continuous system eq. (12) by sampling the continuous input signal $\mathbf{u}(\cdot)$ at time intervals $\Delta t$: $(\mathbf{u}_0, \mathbf{u}_1, \ldots) = (\mathbf{u}(0\Delta t), \mathbf{u}(1\Delta t), \ldots)$. The SSMs store the continuous LTI systems. When evaluating on a discrete input, they discretize the continuous systems using a trainable step size $\Delta t$ and either the Bilinear or the ZOH descritization.

## C   PROOF OF THE TRANSFER FUNCTION DEVIATION

In this section, we prove a result on the difference between the transfer functions $G_{\text{DPLR}}$ and $G_{\text{Diag}}$. The starting point is to use the Woodbury matrix identity to separate out the rank-1 part in the resolvent that appears in $G_{\text{DPLR}}$. In section 3, we let $\mathbf{C} = \mathbf{e}_\ell^\top \mathbf{V}_H$ for some fixed $\ell$. Since we will reserve the letter $\ell$ as an index in the proof, in the appendices, we change the notation and assume $\mathbf{C} = \mathbf{e}_p^\top \mathbf{V}_H$. While this introduces a notation collision with the length of the output vector $\mathbf{y}$, it does not cause any confusion in the proofs.

**Lemma 1.** Let $G_{\text{DPLR}}$ and $G_{\text{Diag}}$ be defined by eq. (8). For any $s \in \mathbb{C}$ with $\text{Re}(s) = 0$, we have

$$G_{\text{DPLR}}(s) - G_{\text{Diag}}(s) = \frac{-s\frac{(-1)^{n-1}\prod_{j=1}^{n-1}(j-s)}{\prod_{j=1}^{n}(j+s)}\sqrt{2p-1}\frac{\prod_{j=0}^{p-2}(s-j)}{\prod_{j=1}^{p}(s+j)}}{\sqrt{2}\left(1 + s\frac{(-1)^{n-1}\prod_{j=1}^{n-1}(j-s)}{\prod_{j=1}^{n}(j+s)}\right)}. \tag{16}$$

The proof of Lemma 1 is technical. Here, we provide an intuitive explanation before diving into the proof. The idea is to expand the term $(s\mathbf{I} - \mathbf{A}_H^\perp + \mathbf{B}_H\mathbf{B}_H^\top/2)^{-1}$ in the expression of $G_{\text{Diag}}$ using the Woodbury matrix identity (Woodbury, 1950), which leads to a primary term $(s\mathbf{I} - \mathbf{A}_H^\perp)^{-1}$ that gets canceled with that in $G_{\text{DPLR}}$ and residual terms that are expanded by Cramer's rule. The importance of eq. (16) is that it reduces the complicated matrix inversions to elementary operations, enabling further analysis.

*Proof of Lemma 1.* For notational cleanliness, in this proof, we define $\mathbf{A} = \mathbf{A}_H$, $\mathbf{A}^\perp = \mathbf{A}_H^\perp$, and $\mathbf{B} = \mathbf{B}_H$. To begin with, we expand $(s\mathbf{I} - \mathbf{A}_H^\perp)^{-1}\mathbf{B}_H$ using the Woodbury matrix identity (Wood-

bury, 1950):

$$
\begin{aligned}
(s\mathbf{I} - \mathbf{A}_H^\perp)^{-1}\mathbf{B}_H &= (s\mathbf{I} - \mathbf{A} - \mathbf{B}\mathbf{B}^\top)^{-1}\mathbf{B} \\
&= \left[ (s\mathbf{I} - \mathbf{A})^{-1} + (s\mathbf{I} - \mathbf{A})^{-1}\mathbf{B}(1 - \mathbf{B}^\top(s\mathbf{I} - \mathbf{A})^{-1}\mathbf{B})^{-1}\mathbf{B}^\top(s\mathbf{I} - \mathbf{A})^{-1} \right]\mathbf{B} \\
&= (s\mathbf{I} - \mathbf{A})^{-1}\mathbf{B} + \frac{\mathbf{B}^\top(s\mathbf{I} - \mathbf{A})^{-1}\mathbf{B}}{1 - \mathbf{B}^\top(s\mathbf{I} - \mathbf{A})^{-1}\mathbf{B}}(s\mathbf{I} - \mathbf{A})^{-1}\mathbf{B} \\
&= (s\mathbf{I} - \mathbf{A})^{-1}\mathbf{B} + \left( 1 + \frac{2\mathbf{B}^\top(s\mathbf{I} - \mathbf{A})^{-1}\mathbf{B} - 1}{1 - \mathbf{B}^\top(s\mathbf{I} - \mathbf{A})^{-1}\mathbf{B}} \right)(s\mathbf{I} - \mathbf{A})^{-1}\mathbf{B} \\
&= 2(s\mathbf{I} - \mathbf{A})^{-1}\mathbf{B} + \frac{2\mathbf{B}^\top(s\mathbf{I} - \mathbf{A})^{-1}\mathbf{B} - 1}{1 - \mathbf{B}^\top(s\mathbf{I} - \mathbf{A})^{-1}\mathbf{B}}(s\mathbf{I} - \mathbf{A})^{-1}\mathbf{B}.
\end{aligned}
$$

Hence, when $\mathbf{C}_{\mathrm{DPLR}} = \mathbf{C}_{\mathrm{Diag}} = \mathbf{I}$, the difference between $G_{\mathrm{DPLR}}$ and $G_{\mathrm{Diag}}$ can be written as

$$
\frac{1}{2}(s\mathbf{I} - \mathbf{A}_H^\perp)^{-1}\mathbf{B}_H - (s\mathbf{I} - \mathbf{A}_H)^{-1}\mathbf{B}_H = \frac{2\mathbf{B}^\top(s\mathbf{I} - \mathbf{A})^{-1}\mathbf{B} - 1}{2 - 2\mathbf{B}^\top(s\mathbf{I} - \mathbf{A})^{-1}\mathbf{B}}(s\mathbf{I} - \mathbf{A})^{-1}\mathbf{B}. \tag{17}
$$

Our next step is to study $\mathbf{B}^\top(s\mathbf{I} - \mathbf{A})^{-1}\mathbf{B}$ that appears in eq. (17). To wit, we use Hua's identity (Cohn, 2003) to obtain

$$
\begin{aligned}
\mathbf{B}^\top(s\mathbf{I} - \mathbf{A})^{-1}\mathbf{B} &= \mathbf{B}^\top \left( -\mathbf{A}^{-1} + \left( \mathbf{A} - \frac{1}{s}\mathbf{A}^2 \right)^{-1} \right)\mathbf{B} \\
&= \mathbf{B}^\top \left( -\mathbf{A}^{-1} + s\left( \mathbf{A}(s\mathbf{I} - \mathbf{A}) \right)^{-1} \right)\mathbf{B}.
\end{aligned}
$$

It is easy to see that $\mathbf{B}^\top\mathbf{A}^{-1}\mathbf{B} = -1/2$. Hence, we have

$$
\mathbf{B}^\top(s\mathbf{I} - \mathbf{A})^{-1}\mathbf{B} = \frac{1}{2} + s\mathbf{B}^\top(s\mathbf{I} - \mathbf{A})^{-1}\mathbf{A}^{-1}\mathbf{B}.
$$

Note that when $s = 0$, the second term in the expression above vanishes, and therefore we already have that $(\mathbf{A}^\perp)^{-1}\mathbf{B}/2 = \mathbf{A}^{-1}\mathbf{B}$. To deal with the general case when $s$ is a purely imaginary number, we first note that $\mathbf{A}^{-1}\mathbf{B} = -\mathbf{e}_1/\sqrt{2}$ because $\mathbf{B}$ is $-1/\sqrt{2}$ times the first column of $\mathbf{A}$. Hence, $s\mathbf{B}^\top(s\mathbf{I} - \mathbf{A})^{-1}\mathbf{A}^{-1}\mathbf{B}$ is equal to $s$ times the first coordinate of $\mathbf{B}^\top(s\mathbf{I} - \mathbf{A})^{-1}$, which we now compute using Cramer's rule. The first coordinate of $\mathbf{B}^\top(s\mathbf{I} - \mathbf{A})^{-1}$ can be written as

$$
\mathbf{B}^\top(s\mathbf{I} - \mathbf{A})^{-1}\mathbf{e}_1 = \overline{\mathbf{e}_1^\top(s\mathbf{I} - \mathbf{A})^{-*}\mathbf{B}} = \overline{\mathbf{e}_1^\top(-s\mathbf{I} - \mathbf{A}^*)^{-1}\mathbf{B}},
$$

where $\bar{s} = -s$ since $s$ is purely imaginary. By Cramer's rule, we have that

$$
\mathbf{e}_1^\top(-s\mathbf{I} - \mathbf{A}^*)^{-1}\mathbf{B} = \frac{\det \begin{bmatrix} 1 & \sqrt{3} & \sqrt{5} & \cdots & \sqrt{2n-1} \\ \sqrt{3} & 2-s & \sqrt{15} & \cdots & \sqrt{3(2n-1)} \\ \sqrt{5} & 0 & 3-s & \cdots & \sqrt{5(2n-1)} \\ \vdots & \vdots & \vdots & \ddots & \vdots \\ \sqrt{2n-1} & 0 & 0 & \cdots & n-s \end{bmatrix}}{\sqrt{2}\det \begin{bmatrix} 1-s & \sqrt{3} & \sqrt{5} & \cdots & \sqrt{2n-1} \\ 0 & 2-s & \sqrt{15} & \cdots & \sqrt{3(2n-1)} \\ 0 & 0 & 3-s & \cdots & \sqrt{5(2n-1)} \\ \vdots & \vdots & \vdots & \ddots & \vdots \\ 0 & 0 & 0 & \cdots & n-s \end{bmatrix}}.
$$

Obviously, the denominator is $\sqrt{2}\prod_{j=1}^n(j-s)$. We compute the numerator by solving a recurrence. We use $D_n$ to denote this determinant. Hence, we have $D_1 = 1$ and $D_2 = -1 - s$. To compute $D_n$, we expand the last row and obtain

$$
D_n = (-1)^{n+1}\sqrt{2n-1}\det \begin{bmatrix} \sqrt{3} & \sqrt{5} & \cdots & \sqrt{2n-3} & \sqrt{2n-1} \\ 2-s & \sqrt{15} & \cdots & \sqrt{3(2n-3)} & \sqrt{3(2n-1)} \\ 0 & 3-s & \cdots & \sqrt{5(2n-3)} & \sqrt{5(2n-1)} \\ \vdots & \vdots & \ddots & \vdots & \vdots \\ 0 & 0 & \cdots & n-1-s & \sqrt{(2n-3)(2n-1)} \end{bmatrix} + (n-s)D_{n-1}.
$$

To compute the determinant of this submatrix, we have

$$
\det\begin{bmatrix}
\sqrt{3} & \sqrt{5} & \cdots & \sqrt{2n-3} & \sqrt{2n-1} \\
2-s & \sqrt{15} & \cdots & \sqrt{3(2n-3)} & \sqrt{3(2n-1)} \\
0 & 3-s & \cdots & \sqrt{5(2n-3)} & \sqrt{5(2n-1)} \\
\vdots & \vdots & \ddots & \vdots & \vdots \\
0 & 0 & \cdots & n-1-s & \sqrt{(2n-3)(2n-1)}
\end{bmatrix}
$$

$$
= (-1)^{n-2}\det\begin{bmatrix}
\sqrt{2n-1} & \sqrt{3} & \sqrt{5} & \cdots & \sqrt{2n-3} \\
\sqrt{3(2n-1)} & 2-s & \sqrt{15} & \cdots & \sqrt{3(2n-3)} \\
\sqrt{5(2n-1)} & 0 & 3-s & \cdots & \sqrt{5(2n-3)} \\
\vdots & \vdots & \vdots & \ddots & \vdots \\
\sqrt{(2n-3)(2n-1)} & 0 & 0 & \cdots & n-1-s
\end{bmatrix}
$$

$$
= (-1)^{n-1}\sqrt{2n-1}\,D_{n-1}.
$$

Hence, combining the two equations above, we obtain the following recurrence:

$$
D_n = -(2n-1)D_{n-1} + (n-s)D_{n-1} = (-n+1-s)D_{n-1}.
$$

It is then easy to show that

$$
D_n = (-1-s)(-2-s)\cdots(-(n-1)-s) = (-1)^{n-1}\prod_{j=1}^{n-1}(j+s).
$$

Putting everything together, we have

$$
\mathbf{B}^\top(s\mathbf{I}-\mathbf{A})^{-1}\mathbf{B} = \frac{1}{2} - s\operatorname{conj}\left(\frac{(-1)^{n-1}\prod_{j=1}^{n-1}(j+s)}{2\prod_{j=1}^{n}(j-s)}\right) = \frac{1}{2} - s\frac{(-1)^{n-1}\prod_{j=1}^{n-1}(j-s)}{2\prod_{j=1}^{n}(j+s)}. \quad (18)
$$

Now, it remains to study the term $(s\mathbf{I}-\mathbf{A})^{-1}\mathbf{B}$ in eq. (17). Since it is a vector of length $n$, we study it component-wise, and the derivation is similar to the one above. To begin with, we fix a component $p$ that we wish to study. Then, by Cramer's rule, we have

$$
\mathbf{e}_p^\top(s\mathbf{I}-\mathbf{A})^{-1}\mathbf{B}
$$

$$
= \frac{\det\begin{bmatrix}
s+1 & 0 & 0 & \cdots & 1 & \cdots & 0 \\
\sqrt{3} & s+2 & 0 & \cdots & \sqrt{3} & \cdots & 0 \\
\sqrt{5} & \sqrt{15} & s+3 & \cdots & \sqrt{5} & \cdots & 0 \\
\sqrt{7} & \sqrt{21} & \sqrt{35} & \cdots & \sqrt{7} & \cdots & 0 \\
\vdots & \vdots & \vdots & \ddots & \vdots & \ddots & \vdots \\
\sqrt{2n-1} & \sqrt{3(2n-1)} & \sqrt{5(2n-1)} & \cdots & \sqrt{2n-1} & \cdots & s+n
\end{bmatrix}}{\sqrt{2}\det\begin{bmatrix}
s+1 & 0 & 0 & \cdots & 0 & \cdots & 0 \\
\sqrt{3} & s+2 & 0 & \cdots & 0 & \cdots & 0 \\
\sqrt{5} & \sqrt{15} & s+3 & \cdots & 0 & \cdots & 0 \\
\sqrt{7} & \sqrt{21} & \sqrt{35} & \cdots & 0 & \cdots & 0 \\
\vdots & \vdots & \vdots & \ddots & \vdots & \ddots & \vdots \\
\sqrt{2n-1} & \sqrt{3(2n-1)} & \sqrt{5(2n-1)} & \cdots & \sqrt{(2p-1)(2n-1)} & \cdots & s+n
\end{bmatrix}}.
$$

Clearly, we have that the denominator is equal to $\sqrt{2}\prod_{j=1}^{n}(j+s)$. To compute the numerator, we first subtract the $p$th column from the first column. This shows that the numerator is equal to

$$
s\det\begin{bmatrix}
s+2 & 0 & \cdots & \sqrt{3} & \cdots & 0 \\
\sqrt{15} & s+3 & \cdots & \sqrt{5} & \cdots & 0 \\
\sqrt{21} & \sqrt{35} & \cdots & \sqrt{7} & \cdots & 0 \\
\vdots & \vdots & \ddots & \vdots & \ddots & \vdots \\
\sqrt{3(2n-1)} & \sqrt{5(2n-1)} & \cdots & \sqrt{2n-1} & \cdots & s+n
\end{bmatrix}. \quad (19)
$$

We can then subtract $\sqrt{3}$ times the $(p-1)$th column of the submatrix from the first column, showing that the numerator is equal to

$$
s(s-1)\det\begin{bmatrix}
s+3 & \cdots & \sqrt{5} & \cdots & 0 \\
\sqrt{35} & \cdots & \sqrt{7} & \cdots & 0 \\
\vdots & \ddots & \vdots & \ddots & \vdots \\
\sqrt{5(2n-1)} & \cdots & \sqrt{2n-1} & \cdots & s+n
\end{bmatrix}.
$$

Continuing in this manner, we have that the numerator is equal to

$$
s(s-1)\cdots(s-p+2)\det\begin{bmatrix}
\sqrt{2p-1} & 0 & 0 & \cdots & 0 \\
\sqrt{2p+1} & s+p+1 & 0 & \cdots & 0 \\
\sqrt{2p+3} & \sqrt{(2p+3)(2p+1)} & s+p+2 & \cdots & 0 \\
\vdots & \vdots & \vdots & \ddots & \vdots \\
\sqrt{2n-1} & \sqrt{(2n-1)(2p+1)} & \sqrt{(2n-1)(2p+2)} & \cdots & s+n
\end{bmatrix}
$$

$$
= \sqrt{2p-1}\,s(s-1)\cdots(s-p+2)(s+p+1)(s+p+2)\cdots(s+n).
$$

Hence, we have

$$
\mathbf{e}_p^\top(s\mathbf{I}-\mathbf{A})^{-1}\mathbf{B} = \frac{\sqrt{2p-1}\prod_{j=0}^{p-2}(s-j)}{\sqrt{2}\prod_{j=1}^{p}(s+j)}. \tag{20}
$$

Note that the expression above does not depend on $n$. Combining eq. (17), (18), (20), when $\mathbf{C}_{\text{DPLR}} = \mathbf{C}_{\text{Diag}} = \mathbf{e}_p^\top$, we have

$$
G_{\text{DPLR}}(s) - G_{\text{Diag}}(s) = \mathbf{e}_p^\top\left[\frac{1}{2}(s\mathbf{I}-\mathbf{A}^\perp)^{-1}\mathbf{B} - (s\mathbf{I}-\mathbf{A})^{-1}\mathbf{B}\right]
$$

$$
= \frac{2\left(\frac{1}{2}-s\frac{(-1)^{n-1}\prod_{j=1}^{n-1}(j-s)}{2\prod_{j=1}^{n}(j+s)}\right)-1}{2-2\left(\frac{1}{2}-s\frac{(-1)^{n-1}\prod_{j=1}^{n-1}(j-s)}{2\prod_{j=1}^{n}(j+s)}\right)}\frac{\sqrt{2p-1}\prod_{j=0}^{p-2}(s-j)}{\sqrt{2}\prod_{j=1}^{p}(s+j)} \tag{21}
$$

$$
= \frac{-s\frac{(-1)^{n-1}\prod_{j=1}^{n-1}(j-s)}{\prod_{j=1}^{n}(j+s)}\sqrt{2p-1}\frac{\prod_{j=0}^{p-2}(s-j)}{\prod_{j=1}^{p}(s+j)}}{\sqrt{2}\left(1+s\frac{(-1)^{n-1}\prod_{j=1}^{n-1}(j-s)}{\prod_{j=1}^{n}(j+s)}\right)}.
$$

This completes the proof of the lemma. $\qquad\square$

## D  PROOF OF THEOREM 2

In this section, we prove Theorem 2. The idea is to locate the last spike in the figure of $G_{\text{Diag}}$ (see Figure 1) and control the height of its peak by lower-bounding the denominator of eq. (16).

*Proof of Theorem 2.* Fix an $n \geq p$. Define $s_n$ by

$$
s_n = \max\left\{s \geq 0 \,\middle|\, A(s) := si\frac{(-1)^{n-1}\prod_{j=1}^{n-1}(j-si)}{\prod_{j=1}^{n}(j+si)}\ \text{is real and}\ \leq 0\right\}. \tag{22}
$$

Note that this set is finite because $A(s) \to 1$ as $s \to \infty$; thus, its supremum is attained. Therefore, we have that

$$
\left|1+s_ni\frac{(-1)^{n-1}\prod_{j=1}^{n-1}(j-s_ni)}{\prod_{j=1}^{n}(j+s_ni)}\right| = 1-\left|s_ni\frac{(-1)^{n-1}\prod_{j=1}^{n-1}(j-s_ni)}{\prod_{j=1}^{n}(j+s_ni)}\right| = \frac{|n+s_ni|-s_n}{|n+s_ni|}. \tag{23}
$$

In what follows, we show that $s_n = \Omega(n^2)$[5] Then, combined with Lemma 1, we have that as $n \to \infty$,

$$|G_{\text{DPLR}}(s_n i) - G_{\text{Diag}}(s_n i)| = \frac{s_n^2 \sqrt{2p-1}}{\sqrt{2} |p-1+s_n i| |p+s_n i| (|n+s_n i|-s_n)} = \Theta(1) \frac{1}{\sqrt{n^2+s_n^2} - s_n}$$

$$= \Theta(1) \frac{\sqrt{n^2+s_n^2} + s_n}{n^2}.$$

If we can show that $s_n = \Omega(n^2)$, then we have that $|G_{\text{DPLR}}(s_n i) - G_{\text{Diag}}(s_n i)| = \Omega(1)$ and does not converge to zero. To this end, we first rewrite the expression into

$$A(s_n) = s_n i \frac{(-1)^{n-1} \prod_{j=1}^{n-1}(j-s_n i)}{\prod_{j=1}^{n}(j+s_n i)} = \frac{s_n i}{n+s_n i} \prod_{j=1}^{n-1} \frac{s_n i - j}{s_n i + j} = \frac{s_n i}{n+s_n i} \exp\left(-i2 \sum_{j=1}^{n-1} \arctan \frac{j}{s_n}\right).$$
(24)

Since $\arctan x = \Theta(x)$ as $x \to 0$, if we assume, for a contradiction, that $s_{n_k} = o(n_k{}^2)$ for a subsequence $s_{n_k}$ of $s_n$, then we must have that

$$\sum_{j=1}^{n_k-1} \arctan \frac{j}{s_{n_k}} - \sum_{j=1}^{n_k-1} \arctan \frac{j}{\max\{n_k, 2s_{n_k}\}} \to \infty \qquad \text{as} \qquad k \to \infty.$$

We pick some index $n_k \geq p$ large enough such that $\sum_{j=1}^{n_k-1} \arctan(j/s_{n_k}) - \sum_{j=1}^{n_k-1} \arctan(j/\max\{n_k, 2s_{n_k}\}) \geq 2\pi$. Hence, as $s$ increases from $s_{n_k}$ to $\max\{n_k, 2s_{n_k}\}$, the angle of the unit imaginary number

$$\exp\left(-i2 \sum_{j=1}^{n_k-1} \arctan \frac{j}{s}\right)$$

changes by at least $4\pi$ whereas the angle of $si/(n+si)$ changes by at most $\pi/2$. Hence, the winding number of the curve

$$\Gamma : s \mapsto \frac{si}{n_k+si} \exp\left(-i2 \sum_{j=1}^{n_k-1} \arctan \frac{j}{s}\right), \qquad s \in [s_{n_k}, \max\{n_k, 2s_{n_k}\}],$$

is non-zero. That is, we must have an $s \in (s_{n_k}, \max\{n_k, 2s_{n_k}\})$ such that the angle of $A(s)$ is equal to $\pi$ modulo $2\pi$, but this is a contradiction because $s_{n_k} < s$. Hence, we have $s_n = \Omega(n^2)$. □

## E  PROOF OF THEOREM 1

In this section, we prove Theorem 1. Since the proof is very involved, we provide some intuition here. In Figure 1, we observe that for a sufficiently large $n$, as $|s|$ increases, the difference between the two transfer functions, $G_{\text{DPLR}} - G_{\text{Diag}}$, goes through three stages. In the first stage (i.e., the pre-spike stage), the large spikes have yet developed. In this stage, as $n$ increases, $|G_{\text{DPLR}} - G_{\text{Diag}}|$ decreases uniformly. In the second stage (i.e., the spike stage), the spikes start to occur. This is the stage in which we do not get uniform convergence. However, by carefully controlling the locations and the total measure of the spikes, we can show that when the Fourier transform of a fixed input function with a sufficient decay is multiplied with $G_{\text{DPLR}} - G_{\text{Diag}}$, its integral on the second stage vanishes linearly as $n \to \infty$. Finally, after the last spike, we enter the third stage (i.e., the post-spike stage). In this stage, $|G_{\text{DPLR}} - G_{\text{Diag}}|$ enjoys rapid decay. In what follows, we carefully analyze the three stages separately to prove Theorem 1.

*Proof of Theorem 1.* Let $\mathbf{u}$ satisfy the assumptions in Theorem 1. Without loss of generality, we assume $\hat{u}(s)$ vanishes on $(-\infty, 0]$ because the argument would be symmetric for a negative $s$. Let

$$H_n(s) = G_{\text{DPLR}}(s) - G_{\text{Diag}}(s) = \frac{-s \frac{(-1)^{n-1} \prod_{j=1}^{n-1}(j-s)}{\prod_{j=1}^{n}(j+s)} \sqrt{2p-1} \frac{\prod_{j=0}^{p-2}(s-j)}{\prod_{j=1}^{p}(s+j)}}{\sqrt{2} \left(1 + s \frac{(-1)^{n-1} \prod_{j=1}^{n-1}(j-s)}{\prod_{j=1}^{n}(j+s)}\right)}.$$

---

[5]We say $f(n) = \Omega(g(n))$ if there exists a constant $C > 0$ such that $f(n) \geq Cg(n)$ for all $n \in \mathbb{N}$.

We set $s_n^{(1)} = cn$, where $c$ is a universal constant determined later on, and

$$s_n^{(2)} = \max \left\{ s \geq 0 \middle| A(s) := si \frac{(-1)^{n-1} \prod_{j=1}^{n-1}(j - si)}{\prod_{j=1}^{n}(j + si)} \text{ is purely imaginary and } \mathrm{Im}(A(s)) \geq 0 \right\}. \tag{25}$$

By the same argument as in the proof of Theorem 2, we have $s_n^{(2)} = \mathcal{O}(n^2)$. To compute the integral of $|H_n(si)\hat{\mathbf{u}}(s)|^2$ on $[0, \infty)$, we do so on each of the three stages, marked by $[0, s_n^{(1)})$, $[s_n^{(1)}, s_n^{(2)})$, and $[s_n^{(2)}, \infty)$, respectively. Since $2p - 1$ is a constant appearing unanimously in $|H_n(si)\hat{\mathbf{u}}(s)|^2$ for all $n$, we absorb it into the asymptotic notations in this proof. Unless otherwise stated, the constants in the asymptotic bounds in this proof are universal constants depending only on $p$; in particular, they do not depend on $n$ or $s$.

**Integrate on the pre-spike stage:** Since

$$\left| s \frac{(-1)^{n-1} \prod_{j=1}^{n-1}(j - s)}{\prod_{j=1}^{n}(j + s)} \right| = \frac{|s|}{|n + s|}$$

and $s = \mathcal{O}(n)$ whenever $0 \leq s \leq s_n^{(1)}$, the denominator of $H_n$ is lower-bounded by a constant independent of $n$. Hence, we have $|H_n(si)| = \mathcal{O}(n^{-1})$ on $[0, s_n^{(1)})$. Using Hölder's inequality, we have

$$\int_0^{s_n^{(1)}} |H_n(si)\hat{\mathbf{u}}(s)|^2 \, ds \leq \|H_n(si)\|_{L^\infty([0, s_n^{(1)}))}^2 \|\hat{\mathbf{u}}(s)\|_{L^2([0, s_n^{(1)}))}^2 = \mathcal{O}(n^{-2}). \tag{26}$$

**Integrate on the post-spike stage:** For $s \geq s_n^{(2)}$, the denominator of $H_n$ is lower-bounded by a constant independent of $n$. Hence, we have $|H_n(si)| = \mathcal{O}(s^{-1})$, where the constant does not depend on $n$. Hence, we have

$$\int_{s_n^{(2)}}^\infty |H_n(si)\hat{\mathbf{u}}(s)|^2 \, ds = \int_{s_n^{(2)}}^\infty \mathcal{O}(s^{-2-2q}) ds = \mathcal{O}(n^{-2-4q}) \tag{27}$$

because $s_n^{(2)} = \mathcal{O}(n^2)$.

**Integrate on the spike stage:** To integrate $|H_n(si)\hat{\mathbf{u}}(s)|^2$ on $[s_n^{(1)}, s_n^{(2)}]$, we first define the angle function by

$$a(s) := \arg \left( \frac{si}{n + si} \right) + 2 \sum_{j=1}^{n-1} \arctan \left( \frac{j}{s} \right) = \arctan \left( \frac{n}{s} \right) + 2 \sum_{j=1}^{n-1} \arctan \left( \frac{j}{s} \right)$$

$$\equiv \arg \left( si \frac{(-1)^{n-1} \prod_{j=1}^{n-1}(j - si)}{\prod_{j=1}^{n}(j + si)} \right) \pmod{2\pi}.$$

The importance of $a(s)$ is that when $a(s)$ is close to $(2k + 1)\pi$ for some integer $k$, we get a spike in the figure of $|H_n|$. We therefore partition the oscillation stage into two parts:

$$S_1 = \{ s \in [s_n^{(1)}, s_n^{(2)}) \mid |a(s) - (2k + 1)\pi| < \pi/4 \text{ for some } k \in \mathbb{N} \}, \qquad S_2 = [s_n^{(1)}, s_n^{(2)}) \setminus S_1.$$

The integral on $S_2$ is studied in the same way as the decay stage:

$$\int_{S_2} |H_n(si)\hat{\mathbf{u}}(s)|^2 \, ds \leq \int_{\mathcal{O}(n)}^{\mathcal{O}(n^2)} \mathcal{O}(s^{-2-2q}) ds = \mathcal{O}(n^{-1-2q}). \tag{28}$$

To study the spikes, we first need to derive a simplified expression of the denominator. Fix an $s \in S_1$. We let

$$\alpha(s) = \min_k |a(s) - (2k + 1)\pi|$$

and

$$d(s) = \left| 1 + si \frac{(-1)^{n-1} \prod_{j=1}^{n-1}(j - si)}{\prod_{j=1}^{n}(j + si)} \right|.$$

Since

$$r(s) := 1 - \left| si\frac{(-1)^{n-1}\prod_{j=1}^{n-1}(j-si)}{\prod_{j=1}^{n}(j+si)} \right| = 1 - \frac{s}{\sqrt{s^2+n^2}},$$

by the cosine law, we have (see Figure 6)

$$\cos\left(\frac{\pi}{2} - \frac{\alpha(s)}{2}\right) = \frac{-d^2 + r(s)^2 + 4\sin^2(\alpha(s)/2)}{4r(s)\sin(\alpha(s)/2)}$$

$$\Rightarrow d(s)^2 = r(s)^2 + 4\sin^2\left(\frac{\alpha(s)}{2}\right) - 4r(s)\sin^2\left(\frac{\alpha(s)}{2}\right) \geq r(s)^2 + \sin^2\left(\frac{\alpha(s)}{2}\right),$$

where the last inequality follows from the fact that $r(s) < 1/2$ for a sufficiently large constant $c$ in the definition of $s_n^{(1)}$.[6] Therefore, we have

$$|H_n(si)\hat{\mathbf{u}}(s)|^2 = \mathcal{O}(s^{-2-2q})\frac{1}{d(s)^2} \leq \mathcal{O}(s^{-2-2q})\frac{1}{r(s)^2 + \alpha(s)^2}, \tag{29}$$

where we used the fact that $x/\pi \leq \sin(x) \leq x$ for all $0 \leq x \leq \pi/2$. Clearly, we have

$$r(s)^2 = \left(\frac{\sqrt{s^2+n^2}-s}{\sqrt{s^2+n^2}}\right)^2 = \left(\frac{s^2+n^2-s^2}{\sqrt{s^2+n^2}(\sqrt{s^2+n^2}+s)}\right)^2 = \mathcal{O}\left(\frac{n^4}{s^4}\right) \tag{30}$$

because $s = \Omega(n)$. To study $\alpha(s)$, we first need to compute $a(s)$. To this end, note that since we assume $s = \Omega(n)$, there exist two universal constants $C_1, C_2 > 0$, independent of $n$, such that

$$C_1\frac{j}{s} \leq \arctan\left(\frac{j}{s}\right) \leq C_2\frac{j}{s}, \qquad 1 \leq j \leq n-1.$$

Hence, we have

$$a(s) = \Theta\left(\frac{n^2}{s}\right).$$

By the intermediate value theorem and monotonicity of $a$, there are $k_n = \mathcal{O}(n)$ frequencies $s_1, \ldots, s_{k_n}$ between $s = s_n^{(1)}$ and $s = s_n^{(2)}$ such that $a(s_j) \equiv \pi \pmod{2\pi}$ for all $1 \leq j \leq k_n$. Each $s_j$ is contained in a connected component $S_1^{(j)} = (\xi_j, \zeta_j)$ of $S_1$ and $S_1 = \bigcup_{j=1}^{k_n} S_1^{(j)}$. That is, we have

$$s_n^{(1)} < \xi_{k_n} < s_{k_n} < \zeta_{k_n} < \xi_{k_n-1} < s_{k_n-1} < \zeta_{k_n-1} < \cdots < \xi_1 < s_1 < \zeta_1 < s_n^{(2)}.$$

Moreover, there are two universal constants $C_1, C_2 > 0$, independent of $n$ or $j$, such that

$$C_1 j^{-1}n^2 \leq s_j \leq C_2 j^{-1}n^2.$$

Combined with eq. (30), we have

$$r(s)^2 = \mathcal{O}\left(\frac{j^4}{n^4}\right), \qquad s \in S_1^{(j)}, \quad 1 \leq j \leq k_n,$$

where the constant is universal and does not depend on $n$ or $j$. To integrate $|H_n(si)\hat{\mathbf{u}}(s)|^2$ on $S_1$, we integrate it on each of $(\xi_j, \zeta_j)$. To do so, we study the value of $\alpha(s)$ using the Mean Value Theorem. First, we note that for any given $s_j$, we have

$$\frac{d}{ds}a(s_j) = -\Theta(1)\sum_{k=1}^{n-1}\frac{1}{1+\frac{k^2}{s_j^2}}\frac{k}{s_j^2} = -\Theta(1)\frac{1}{s_j^2}\sum_{k=1}^{n-1}\frac{k}{\frac{s_j^2+k^2}{s_j^2}}$$

$$= -\Theta(1)\sum_{k=1}^{n-1}\frac{k}{s_j^2} = -\Theta(1)\frac{n^2}{s_j^2} = -\Theta(1)\frac{j^2}{n^2},$$

---

[6]This is our only requirement of the universal constant $c$ appearing in the definition of $s_n^{(1)}$.

where the constant in the $\Theta$-notation does not depend on $n$ or $j$. Hence, fixing a $1 \leq j \leq k_n$ and choosing $s \in (\xi_j, \zeta_j)$, by the Mean Value Theorem, we have

$$\alpha(s) = |a(s) - a(s_j)| = \Theta(1)\frac{j^2}{n^2}|s - s_j|.$$

This shows $\zeta_j - s_j, s_j - \xi_j = \Theta(n^2/j^2)$. Hence, we have

$$\int_{\xi_j}^{\zeta_j} |H_n(si)\hat{\mathbf{u}}(s)|^2 \, ds = \mathcal{O}((j^{-1}n^2)^{-2-2q}) \int_{\xi_j}^{\zeta_j} \frac{1}{r(s)^2 + \alpha(s)^2} ds$$

$$\leq \mathcal{O}(j^{2+2q}n^{-4-4q}) \left( \int_{s_j-1}^{s_j+1} \frac{1}{r(s)^2} ds + \int_{\xi_j}^{s_j-1} \frac{1}{\alpha(s)^2} ds + \int_{s_j+1}^{\zeta_j} \frac{1}{\alpha(s)^2} ds \right) \tag{31}$$

$$\leq \mathcal{O}(j^{2+2q}n^{-4-4q}) \left( \frac{n^4}{j^4} + \frac{n^4}{j^4} \int_1^{\Theta(n^2/j^2)} \delta^{-2} d\delta \right) = \mathcal{O}(j^{-2+2q}n^{-4q}).$$

Suppose $q > 1/2$ and let $q' = q - 1/2$. Then, we have

$$\int_{S_1} |H_n(si)\hat{\mathbf{u}}(s)|^2 \, ds = \sum_{j=1}^{k_n} \int_{\xi_j}^{\zeta_j} |H_n(si)\hat{\mathbf{u}}(s)|^2 \, ds = \mathcal{O}(1) \sum_{j=1}^{k_n} j^{-1+2q'} n^{-2-4q'}$$

$$\leq \mathcal{O}(n^{-2}) \sum_{j=1}^{k_n} j^{-1-2q'} = \mathcal{O}(n^{-2}), \tag{32}$$

where the constant in the last $\mathcal{O}$-notation only depends on $p$. Combining eq. (28) and (32), we have that when $q > 1/2$, it holds that

$$\int_{s_n^{(1)}}^{s_n^{(2)}} |H_n(si)\hat{\mathbf{u}}(s)|^2 \, ds = \mathcal{O}(n^{-2}). \tag{33}$$

**Put everything together:** Combining eq. (26), (27), and (33) and applying Parseval's identity, we obtain

$$\|\mathbf{y}_{\text{DPLR}} - \mathbf{y}_{\text{Diag}}\|_{L^2} = \|\hat{\mathbf{y}}_{\text{DPLR}} - \hat{\mathbf{y}}_{\text{Diag}}\|_{L^2} = \|H_n(si)\hat{\mathbf{u}}(s)\|_{L^2}$$

$$= \sqrt{\int_0^{s_n^{(1)}} |H_n(si)\hat{\mathbf{u}}(s)|^2 \, ds + \int_{s_n^{(1)}}^{s_n^{(2)}} |H_n(si)\hat{\mathbf{u}}(s)|^2 \, ds + \int_{s_n^{(2)}}^{\infty} |H_n(si)\hat{\mathbf{u}}(s)|^2 \, ds} = \mathcal{O}(n^{-1}).$$

This completes the proof. □

## F   NUMERICAL EXPERIMENTS ON THEOREM 1 AND 2

In this section, we explain details and show supplementary results for the experiment in section 3 (see Figure 2). The first experiment examines the behaviors of the DPLR system and the diagonal system given a single Fourier mode as an input. By doing so, we observe the "numerical unstable modes" of the S4D model. This corroborates Theorem 2. Then, we compare the two systems using two different input functions: an exponentially decaying function and the unit impulse. We will show that the smoothness condition in Theorem 1 is necessary and the linear convergence rate is tight.

In each of these experiments, we simulate LTI systems on some continuous input signals. It is done as follows: given an input signal $u(t)$[7] and an LTI system $\Sigma$, we fix the step size to be $\Delta t = 10^{-3}$. For some final time step $N$, we discretize our input function to obtain a vector $\mathbf{u} = (u(0), u(\Delta t), \ldots, u(N\Delta t))$. We then discretize the LTI system bilinearly (see Appendix B) and compute its output $\mathbf{y}$ on the input $\mathbf{u}$. We call this procedure "simulate", i.e.,

$$\mathbf{y} = \texttt{simulate}(u, \Sigma, N).$$

In this section, we let $\Sigma_{\text{DPLR},n}$ to be the DPLR system with state size $n$ of S4 and $\Sigma_{\text{Diag},n}$ to be the diagonal system with state size $n$ of S4D, where we always take $\mathbf{C} = \mathbf{e}_1$ and $\mathbf{D} = \mathbf{0}$.

---

[7]Since $u$ will be scalar-valued in this section, we do not make it boldface.

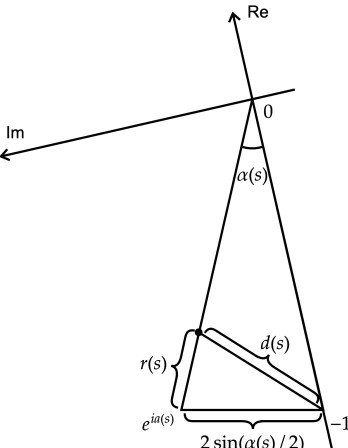

**Figure 6:** Illustration of the proof of Theorem 1.

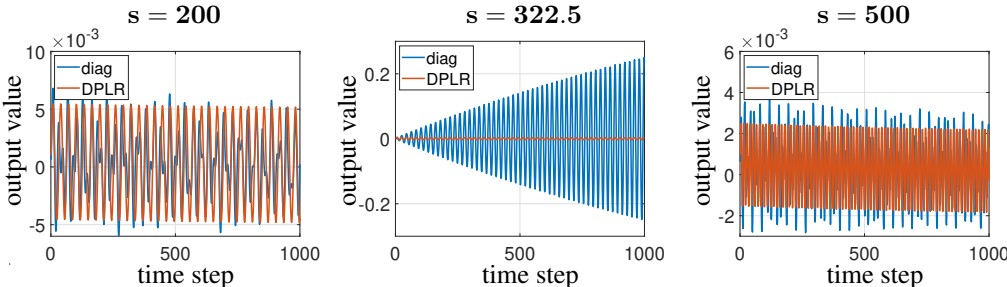

**Figure 7:** Simulated outputs of the DPLR and diagonal systems with cosine wave inputs of different frequencies $s$. Note the scale of the $y$-axis when $s = 322.5$.

### F.1   THE DIAGONAL SYSTEM BEHAVES DIFFERENTLY FOR DISTINCT FOURIER MODES

Our first experiment considers the outputs of $\Sigma_{\text{DPLR},n}$ and $\Sigma_{\text{Diag},n}$ when the input is a cosine wave

$$u_s(t) = \cos(st).$$

This function has a dense Fourier mode at frequency $s$. We fix $n = 32$ and let $s$ change. In Figure 7, we plot $\texttt{simulate}(u_s, \Sigma_{\text{DPLR},32}, 10^3)$ and $\texttt{simulate}(u_s, \Sigma_{\text{Diag},32}, 10^3)$ with $s = 200, 322.5$, and $500$, respectively. We see that when $s = 200$ or $500$, the outputs of the two systems are close to each other - at least, they are on the same order of magnitude. However, when $s = 322.5$, the output of the diagonal system blows up. In fact, this value of $s$ is exactly where the spike in the plot of $\|G_{\text{Diag}}\|$ occurs when $n = 32$. Hence, we visualize the counter-example that shows the divergence in Theorem 2.

### F.2   THE DPLR AND DIAGONAL SYSTEMS CONVERGE ON THE EXPONENTIALLY DECAYING FUNCTION

To test the function-wise convergence of the diagonal system to the DPLR system (see Theorem 1), we consider the following exponentially decaying function:

$$u_e(t) = e^{-t}H(t),$$

where $H = \mathbb{1}_{[0,\infty)}$ is the Heaviside function. The Fourier transform of this function is

$$\hat{u}_e(s) = \frac{1}{1 + is}.$$

Hence, it is a function that satisfies the assumptions of Theorem 1. In the left panel of Figure 8, we show the difference between the two simulated outputs $\|\texttt{simulate}(u_e, \Sigma_{\text{DPLR},n}, 10^4) -$

$\texttt{simulate}(u_e, \Sigma_{\mathrm{Diag},n}, 10^4)\|$ as $n$ increases. We see that as $n$ increases, $\texttt{simulate}(u_e, \Sigma_{\mathrm{Diag},n}$ converges to $\texttt{simulate}(u_e, \Sigma_{\mathrm{DPLR},n}, 10^4)$. Moreover, the slope of the curve is roughly $-1$, indicating a linearly convergence rate as $n \to \infty$. This matches the theoretical statement in Theorem 1.

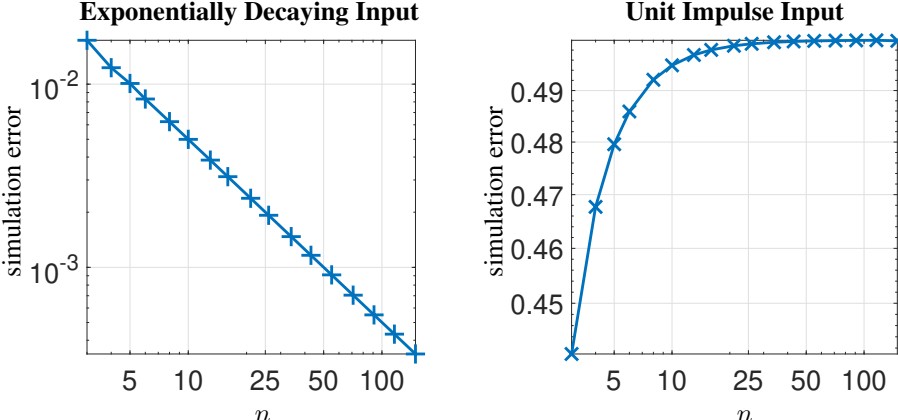

**Figure 8:** The difference between the outputs $\|\texttt{simulate}(u, \Sigma_{\mathrm{DPLR},n}, 10^4) - \texttt{simulate}(u, \Sigma_{\mathrm{Diag},n}, 10^4)\|$ for difference values of $n$ when $u$ is the exponentially decaying function $u_e$ (left) and the unit impulse signal $\delta_0$ (right).

In Figure 2, we show the behaviors of the two simulated outputs as $n$ increases. For the exponentially decaying input function $u_e$, the outputs demonstrate a clear pattern of convergence as $n \to \infty$.

### F.3 THE DPLR AND DIAGONAL SYSTEMS DIVERGE ON THE UNIT IMPULSE

Our experiment with the exponentially decaying input shows that the DPLR and diagonal systems converge on a sufficiently smooth input function. One may wonder, however, if the smoothness condition is necessary. To show that a mild one is indeed required, we consider the Dirac delta function $\delta_0$. It is well-known that the Fourier transform of it is constantly one:

$$\hat{\delta}_0(s) = 1, \qquad s \in \mathbb{R}.$$

In that sense, $\delta_0$ is highly non-smooth as its Fourier transform does not decay at all. Since $\delta_0$ is a distribution rather than a classical function, we cannot sample it directly. However, we can mimic it by setting the discrete input to be the unit impulse $(1, 0, 0, \ldots, 0)$. In the right panel of Figure 8, we see that $\|\texttt{simulate}(\delta_0, \Sigma_{\mathrm{DPLR},n}, 10^4) - \texttt{simulate}(\delta_0, \Sigma_{\mathrm{Diag},n}, 10^4)\|$ does not decay as $n$ increases. We can take a closer look in Figure 2, where we plot the two output functions with different state-space dimensions $n$. In particular, we see that as $n$ increases, the output of the DPLR system remains the same, whereas the output of the diagonal system becomes more oscillatory. We do not have convergence. The oscillatory behavior can be explained by our observation in Figure 1: the larger the $n$, the later the spike emerges. This means that for a larger $n$, the outputs of two systems differ at a higher frequency (i.e., a more oscillatory mode).

### G IMPLICATION OF THE THEORY: NON-ROBUSTNESS OF THE DIAGONAL INITIALIZATION

The analysis of $G_{\mathrm{DPLR}}$ and $G_{\mathrm{Diag}}$ suggests the following caveat: while the S4 and the S4D models tend to pertain similar behaviors as $n$ gets large, the diagonal initialization scheme used by the S4D model is less robust to perturbations in the frequency domain (see Figure 1). In particular, by eq. (3), for a fixed state size $n$, input signals with frequency modes dense at the spikes of the plot of $|G_{\mathrm{Diag}}|$ are harder to process for the SSM. In turn, the S4D model is unstable near these modes. This does not happen with the S4 model. Our observation suggests that instead of replacing the ill-posed diagonalization problem with a well-conditioned but distinct one (i.e., the S4D initialization), which creates a large backward error $|G_{\mathrm{DPLR}}(s) - G_{\mathrm{Diag}}(s)|$, one should solve the ill-posed problem using a backward stable algorithm, even if the forward error (i.e., the miscalculation of eigenvalues and eigenvectors) will be large (see section 4).

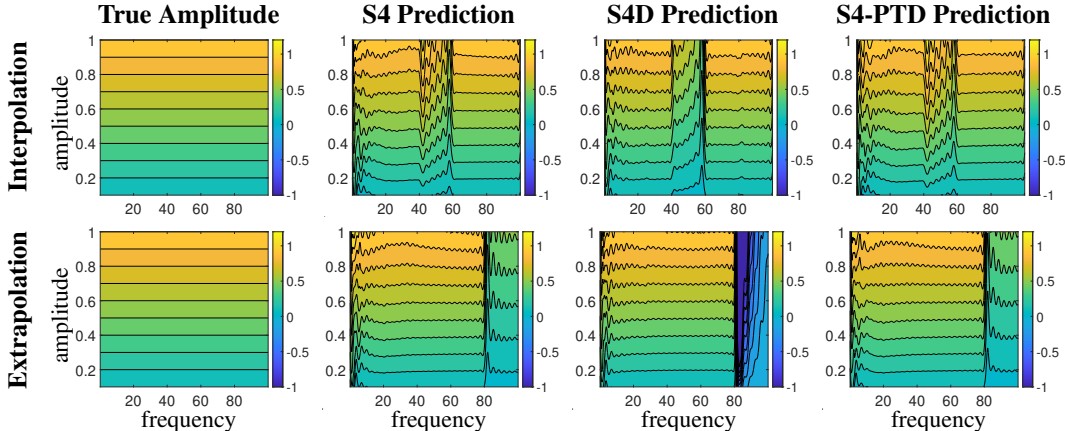

**Figure 9:** An SSM is trained to learn the amplitude of a signal $A\sin(st)$ given its samples. The model is tested using signals $\{A_j\sin(s_jt)\}$, where $A_j$ and $s_j$ are sampled on a uniform grid on $[0,1]\times[0,100]$. For each $A_j$ and $s_j$, the model predicts an amplitude $\tilde{A}_j$, which we show in the contour plots. The top row shows the interpolation result, where the functions in the training datasets have frequencies only in $s\in[0,40]\cup[60,100]$. The bottom row shows the extrapolation result, where the functions in the training datasets have frequencies only in $s\in[0,80]$. The figure shows the interpolation and extrapolation results for the S4 model, the S4D model, and the S4-PTD model (see section 4). We observe that our S4-PTD model interpolates and extrapolates better than the S4D model. In particular, the S4D model is not stable around $s=80$, where the predicted amplitude decreases to $-4$ when the true value increases from 0 to 1. More quantitative results for the interpolation and extrapolation errors can be found in Appendix G.

We demonstrate on a synthetic example that the S4D model, regardless of its size $n$, is not robust under input perturbation of certain frequency modes (which depend on $n$). Our training set contains sinusoidal signals parameterized by a frequency $s$ and an amplitude $A$:

$$\mathbf{u}_j(t) = A_j\sin(s_jt),$$

where $A_j\in[0,1]$ and $s_j\in S_{\text{interp}}:=[0,40]\cup[60,100]$ for an interpolation problem or $s_j\in S_{\text{extrap}}:=[0,80]$ for an extrapolation problem. We sample each input function $\mathbf{u}_j(\cdot)$ uniformly on $t\in[0,10^4]$ and train an S4 model and an S4D model with $n=32$, respectively. Our goal is to learn $s$ and $A$ from the sequential input. In Figure 9, we plot the model prediction of the amplitude $A$ over a test set of signals for which $s$ and $A$ are on a uniform grid on $[0,100]\times[0,1]$.

Figure 9 shows that while both the S4 and S4D models predict well on sampled domains (i.e., $S_{\text{interp}}$ and $S_{\text{extrap}}$), the S4 model is significantly better at interpolating and extrapolating on the unsampled domains. In particular, the S4D model suffers from an extrapolation disaster: for $s>80$, as the true amplitude of the signal increases from 0 to 1, the predicted amplitude decreases monotonically with a minimum value less than $-4$. This happens because $|G_{\text{Diag}}|$ has a spike around $|s|=83$, making the information of $s\in[0,80]$ impossible to transfer to $s\in[80,100]$. Hence, while the S4D initialization stabilizes the diagonalization process, making the computation more efficient, its underlying state-space model is unstable near certain Fourier modes, impairing its robustness (see also section 5.2).

We quantitatively evaluate this point in Figure 10. More specifically, we define four domains by

$$S_{\text{interp}} = [0,40]\cup[60,100], \qquad S_{\text{extrap}} = [0,80],$$
$$U_{\text{interp}} = [0,100]\setminus S_{\text{interp}} = (40,60), \qquad U_{\text{extrap}} = [0,100]\setminus S_{\text{extrap}} = (80,100).$$

We are given training samples only with $s\in S_{\text{interp}}$ (resp. $s\in S_{\text{extrap}}$) and we test the sequential model on the entire domain $[0,100]$. Given a test set, we measure the mean-squared error of our model. We uniformly sample the test set from $s\in S_{\text{interp}}$ (resp. $s\in S_{\text{extrap}}$) and $s\in U_{\text{interp}}$ (resp. $s\in U_{\text{extrap}}$) to evaluate our models' performance on generalization to unseen data in the seen domain, and their performance on interpolation (resp. extrapolation). We see that the S4D model performs even better on the seen domain (i.e., $S_{\text{interp}}$ or $S_{\text{extrap}}$), but its interpolation and extrapolation capabilities are much worse than those of the S4 and our S4-PTD models.

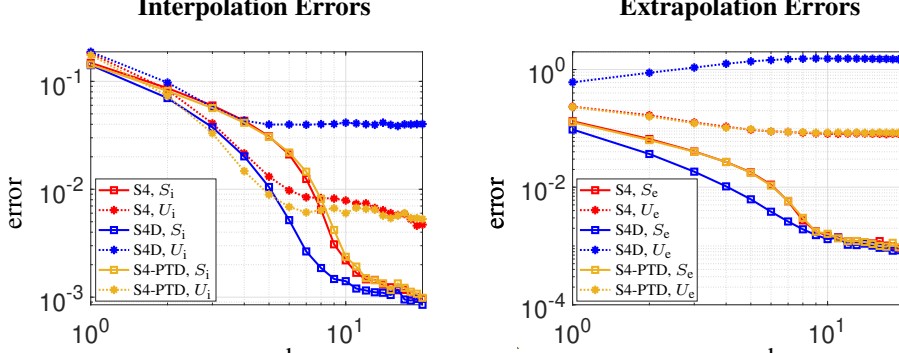

**Figure 10:** The interpolation and extrapolation errors of predicting the amplitude of a sinusoidal signal made by the S4, S4D, and S4-PTD models. Each curve shows the mean-squared test error of one model on either the seen domain, $S_i$ or $S_e$, or the unseen domain, $U_i$ or $U_e$. The yellow curve for the S4-PTD model and the red curve for the S4 model almost overlap in the extrapolation problem.

## H    PROOF OF RESULTS IN SECTION 4

In this section, we present the proof of Theorem 3. In addition, we introduce a probabilistic statement of the eigenvector condition number of a matrix perturbed by a random Gaussian matrix.

The proof of Theorem 3 is a classical forward error analysis, but to maintain the best result, we need to explicitly compute the resolvent of $\mathbf{A}_H$.

*Proof of Theorem 3.* For notational cleanliness, in this proof, we define $\mathbf{A} = \mathbf{A}_H$, $\mathbf{B} = \mathbf{B}_H$, and $\mathbf{C} = \mathbf{C}_{\mathrm{DPLR}}\mathbf{V}_H^{-1}$. We have

$$
\begin{aligned}
|G_{\mathrm{Pert}}(s) - G_{\mathrm{DPLR}}(s)| &= \left| \mathbf{B}(s\mathbf{I} - \mathbf{A})^{-1}\mathbf{C} - \mathbf{B}(s\mathbf{I} - \mathbf{A} - \mathbf{E})^{-1}\mathbf{C} \right| \\
&= \left| \mathbf{B}\big((s\mathbf{I} - \mathbf{A})^{-1} - (s\mathbf{I} - \mathbf{A} - \mathbf{E})^{-1}\big)\mathbf{C} \right| \\
&\leq \|\mathbf{B}\|_2 \left\| (s\mathbf{I} - \mathbf{A})^{-1} - (s\mathbf{I} - \mathbf{A} - \mathbf{E})^{-1} \right\|_2 \|\mathbf{C}\|_2,
\end{aligned}
$$

where, by a result in Demmel (1992), we have

$$
\left\| (s\mathbf{I} - \mathbf{A})^{-1} - (s\mathbf{I} - \mathbf{A} - \mathbf{E})^{-1} \right\|_2 \leq \|\mathbf{E}\|_2 \left\| (s\mathbf{I} - \mathbf{A})^{-1} \right\|_2^2 + \mathcal{O}(\|\mathbf{E}\|_2^2) \left\| (s\mathbf{I} - \mathbf{A})^{-1} \right\|_2. \quad (34)
$$

We set

$$
\begin{bmatrix}
c_{1,1} & 0 & 0 & \cdots & 0 \\
c_{2,1} & c_{2,2} & 0 & \cdots & 0 \\
c_{3,1} & c_{3,2} & c_{3,3} & \cdots & 0 \\
\vdots & \vdots & \vdots & \ddots & \vdots \\
c_{n,1} & c_{n,2} & c_{n,3} & \cdots & c_{n,n}
\end{bmatrix}
= (-s\mathbf{I} + \mathbf{A})^{-1} =
\begin{bmatrix}
1-s & 0 & 0 & \cdots & 0 \\
\sqrt{3} & 2-s & 0 & \cdots & 0 \\
\sqrt{5} & \sqrt{15} & 3-s & \cdots & 0 \\
\vdots & \vdots & \vdots & \ddots & \vdots \\
\sqrt{2n-1} & \sqrt{3(2n-1)} & \sqrt{5(2n-1)} & \cdots & n-s
\end{bmatrix}^{-1}.
$$

Then, fixing a column $i$ and a row $j \geq i$, we have

$$
\begin{cases}
\displaystyle\sum_{k=i}^{j-1} c_{k,i}\sqrt{2j-1}\sqrt{2k-1} + c_{j,i}(j-s) = 0, & (35) \\[4mm]
\displaystyle\sum_{k=i}^{j} c_{k,i}\sqrt{2j+1}\sqrt{2k-1} + c_{j+1,i}(j+1-s) = 0. & (36)
\end{cases}
$$

Multiplying eq. (35) by $\sqrt{2j+1}/\sqrt{2j-1}$, we have

$$
\sum_{k=i}^{j-1} c_{k,i}\sqrt{2j+1}\sqrt{2k-1} + \frac{\sqrt{2j+1}}{\sqrt{2j-1}}c_{j,i}(j-s) = 0. \quad (37)
$$

Subtracting eq. (37) from eq. (35), we have

$$c_{j,i}\sqrt{2j+1}\sqrt{2j-1} - c_{j,i}\frac{\sqrt{2j+1}}{\sqrt{2j-1}}(j-s) + c_{j+1,i}(j+1-s) = 0.$$

After simplifying, we get the recurrence relation

$$c_{i,i} = \frac{1}{i-s}, \qquad c_{i+1,i} = -\frac{\sqrt{2i-1}\sqrt{2i+1}}{(i-s)(i+1-s)},$$

$$c_{j+1,i} = -\frac{(j+s-1)\sqrt{2j+1}}{(j-s+1)\sqrt{2j-1}}c_{j,i}, \qquad j \geq i+1.$$

Solving this recurrence relation gives us

$$c_{k,i} = (-1)^{k-i}\frac{\sqrt{2i-1}\sqrt{2i+1}}{(i-s)(i+1-s)}\frac{\sqrt{2k-1}}{\sqrt{2i+1}}\frac{\prod_{\ell=i}^{k-2}(\ell+s)}{\prod_{\ell=i+2}^{k}(\ell-s)}, \qquad k \geq i+1.$$

Since $s$ is purely imaginary, we have

$$\left|\frac{\ell+s}{\ell-s}\right| = 1.$$

Therefore, we can control the size of $c_{k,i}$ by[8]

$$|c_{k,i}| = \frac{\sqrt{2i-1}\sqrt{2k-1}}{|i-s||i+1-s|}\frac{|i+s||i+1+s|}{|k-1-s||k-s|} = \frac{\sqrt{2i-1}\sqrt{2k-1}}{|k-1-s||k-s|}, \qquad k \geq i+2.$$

Clearly, this value is maximized when $s = 0$. Hence, we have

$$|c_{k,i}|^2 \leq \frac{(2i-1)(2k-1)}{(k-1)^2 k^2} \leq \frac{4i}{(k-1)^2 k}.$$

Note that this inequality holds also for the case when $k = i+1$. Now, we have

$$\left\|(s\mathbf{I}-\mathbf{A})^{-1}\right\|_2^2 \leq \left\|(s\mathbf{I}-\mathbf{A})^{-1}\right\|_F^2 \leq \sum_{k=2}^{n}\sum_{i=1}^{k-1}\frac{4i}{(k-1)^2 k} + \sum_{i=1}^{n}\frac{1}{i^2}$$

$$\leq \sum_{k=2}^{n}\frac{2(k-1)k}{(k-1)^2 k} + 2 \leq 2\ln(n) + 4.$$

The result follows from eq. (34). □

In section 4, we show the effect of a "best-case" perturbation scheme on the eigenvector condition number. In this appendix, we present a probabilistic statement of the eigenvector condition number in the "average case." Our result is heavily based on Banks et al. (2021, Thm. 1.5). To simplify our statement, Given a square matrix $\mathbf{M}$, we define its eigenvector condition number to be

$$\kappa_{\text{eig}}(\mathbf{M}) = \inf_{\mathbf{V}} \kappa(\mathbf{V}),$$

where $\mathbf{V}$ ranges over all invertible matrices such that $\mathbf{M} = \mathbf{V}\mathbf{\Lambda}\mathbf{V}^{-1}$ for some diagonal $\mathbf{\Lambda}$.

**Theorem 5.** Given any matrix $\mathbf{A} \in \mathbb{C}^{n \times n}$, perturbation size $\epsilon \in (0,1)$, and spectral radius $R > 0$. Let $\mathbf{G}_n \in \mathbb{C}^{n \times n}$ be the Ginibre matrix and let $\Omega$ be the event that the spectrum of $\mathbf{A} + \epsilon\mathbf{G}_n$ is contained in $D_R(0)$, the disk centered at zero of radius $R$. Then, we have

$$\mathbb{E}\left[\kappa_{\text{eig}}(\mathbf{A} + \epsilon\mathbf{G}_n)^2 | \Omega\right] \leq \|\mathbf{A}\|^2 \frac{R^2 n^3}{\epsilon^2 \mathbb{P}(\Omega)}.$$

*Proof.* By Banks et al. (2021, Thm. 1.5), we have that

$$\mathbb{E}\left[\sum_{j=1}^{n}\kappa(\lambda_i)^2 \mathbb{1}_{\{\lambda_i \in D_R(0)\}}\right] \leq \|\mathbf{A}\|^2 \frac{R^2 n^2}{\epsilon^2},$$

---

[8]With a slight abuse of notation, the letter $i$ here stands for a real-valued index instead of the imaginary unit.

where $\lambda_1, \ldots, \lambda_n$ are eigenvalues of $\mathbf{A} + \epsilon\mathbf{G}_n$ and $\kappa(\lambda_i)$ is defined in Banks et al. (2021). When $\lambda_j \in D_R(0)$ for all $1 \leq j \leq n$, we have

$$\kappa_{\text{eig}}(\mathbf{A} + \epsilon\mathbf{G}_n)^2 \leq n \sum_{j=1}^{n} \kappa(\lambda_i)^2.$$

Hence, this shows

$$\frac{1}{n}\mathbb{E}\left[\kappa_{\text{eig}}(\mathbf{A} + \epsilon\mathbf{G}_n)^2 | \Omega\right]\mathbb{P}(\Omega) + \mathbb{E}\left[\sum_{j=1}^{n}\kappa(\lambda_i)^2\mathbb{1}_{\{\lambda_i \in D_R(0)\}}\middle|\Omega^C\right]\mathbb{P}(\Omega^C) \leq \|\mathbf{A}\|^2\frac{R^2 n^2}{\epsilon^2}.$$

We are done. $\qquad\qquad\qquad\qquad\qquad\qquad\qquad\qquad\qquad\qquad\qquad\qquad\qquad\qquad\qquad\qquad\qquad\square$

Comparing Theorem 5 to Theorem 4, we note that the bound in Theorem 4 is slightly better than that in Theorem 5. However, the Gaussian perturbation in Theorem 5 is problem-independent and can be generically implemented, whereas it is not necessarily easy to identify the perturbation in Theorem 4.

## I  NUMERICAL EXPERIMENTS ON THEOREM 3 AND 4

### I.1  THE RELATIONSHIP BETWEEN $\|\mathbf{E}\|$ AND THE TRANSFER FUNCTION PERTURBATION

In the proof of Theorem 3, we used the inequality between the matrix spectral norm and the Frobenius norm:
$$\left\|(s\mathbf{I} - \mathbf{A})^{-1}\right\|_2^2 \leq \left\|(s\mathbf{I} - \mathbf{A})^{-1}\right\|_F^2.$$
In practice, this estimate is rarely sharp, given that $s\mathbf{I} - \mathbf{A}$ is a dense matrix. Another non-sharpness in the average case comes from the inequality

$$\left|\mathbf{B}\big((s\mathbf{I} - \mathbf{A})^{-1} - (s\mathbf{I} - \mathbf{A} - \mathbf{E})^{-1}\big)\mathbf{C}\right| \leq \|\mathbf{B}\|_2 \left\|(s\mathbf{I} - \mathbf{A})^{-1} - (s\mathbf{I} - \mathbf{A} - \mathbf{E})^{-1}\right\|_2 \|\mathbf{C}\|_2.$$

To understand the average-case perturbation of the transfer function, we conduct a simulation, where we sample $\mathbf{E}$ and $\mathbf{C}$ randomly but restrict that $\|\mathbf{E}\| = 0.1$ and $\|\mathbf{C}\| = 1$. We then compute the maximum error between the perturbed transfer function and the unperturbed one. In Figure 11, we observe that as $n$ increases, instead of increasing logarithmically, the maximum error $\|G_{\text{DPLR}} - G_{\text{Pert}}\|_\infty$ averaged over all trials decays quadratically, i.e., $\|G_{\text{DPLR}} - G_{\text{Pert}}\|_\infty = \mathcal{O}(n^{-2})$. Hence, in the average case, we obtain a better empirical error estimate compared to the worst-case error estimate in Theorem 3. We remark, however, that $\|G_{\text{DPLR}} - G_{\text{Pert}}\|_\infty$ is not a relative error because while we fix $\|\mathbf{E}\| \leq \epsilon$, the norm of the state matrix $\|\mathbf{A}_H\|$ increases as $n \to \infty$.

### I.2  THE RELATIONSHIP BETWEEN $\|\mathbf{E}\|$ AND $\kappa_{\text{EIG}}(\tilde{\mathbf{A}}_H)$

The performance of our perturbed model is heavily based on two things: the perturbation size $\|\mathbf{E}\|$ and the condition number of $\tilde{\mathbf{V}}_H$. The former value controls the difference between our initialization to the known-to-be-good HiPPO initialization, whereas the latter one controls the unfairness when transforming the states via $\tilde{\mathbf{V}}_H$. By Theorem 4, the condition number $\kappa(\tilde{\mathbf{V}}_H)$ should depend linearly on $\|E\|^{-1}$ and depend sub-quadratically on $n$, the state space size.

In this section, we present a numerical experiment that investigates the relationship between these three values. To do so, we solve the optimization problem in eq. (11) with different state space dimensions $n$ and values of $\gamma$. We then record the size of the perturbation and the eigenvector condition number of the perturbed matrix. In Figure 12, we see that the eigenvector condition number $\kappa_{\text{eig}}(\tilde{\mathbf{A}}_H)$ depends polynomially on both the state space dimension $n$ and the relative perturbation size $\|\mathbf{E}\|/\|\mathbf{A}_H\|$. Numerical values are reported in Table 2 and 3. Using the data, one can compute that we have $\kappa_{\text{eig}}(\tilde{\mathbf{A}}_H) = \mathcal{O}((\|\mathbf{E}\|/\|\mathbf{A}_H\|)^{-p})$, where $p \approx 0.87$. Hence, we are doing slightly better than the theory of Theorem 4. Another surprising observation that can be made with a little bit computation is that if we normalize $\mathbf{A}_H$ to have a spectral norm of 1, then the eigenvector condition number $\kappa_{\text{eig}}(\tilde{\mathbf{A}}_H)$ does not depend on $n$ at all. This is much better than the bound proposed in Theorem 4.

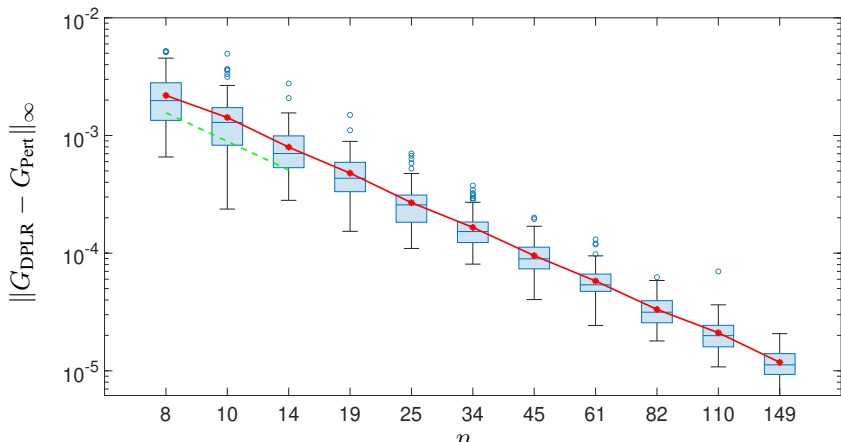

**Figure 11:** The perturbation error $\|G_{\text{DPLR}} - G_{\text{Pert}}\|_\infty$ for different state-space dimension $n$. For each $n$, the matrix $\mathbf{A}_H$ is perturbed by 100 randomly sampled matrices $\mathbf{E}$, respectively. The red curve shows the average error among the 100 trials. The horizontal axis is on the logarithmic scale and the green reference line has a slope of $-2$.

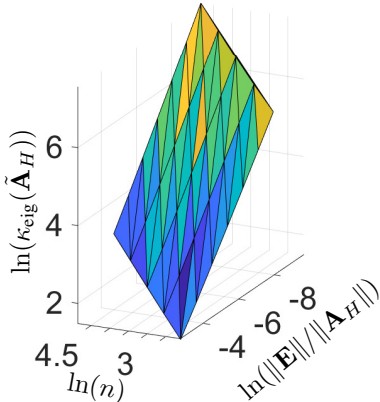

**Figure 12:** The relationship among the state space dimension $n$, the relative perturbation size $\|\mathbf{E}\|/\|\mathbf{A}_H\|$, and the eigenvector condition number $\kappa_{\text{eig}}(\tilde{\mathbf{A}}_H)$.

## J   DETAILS OF EXPERIMENTS IN SECTION 5

In this section, we provide the details of the experiments presented in section 5.

| $n$ \ $\gamma$ | 10 | $10^2$ | $10^3$ | $10^4$ | $10^5$ | $10^6$ | $10^7$ |
|---|---|---|---|---|---|---|---|
| 8 | 4.40e0 | 8.62e0 | 1.73e1 | 3.51e1 | 7.12e1 | 1.45e2 | 2.96e2 |
| 16 | 6.59e0 | 1.32e1 | 2.69e1 | 5.53e1 | 1.14e2 | 2.35e2 | 4.86e2 |
| 32 | 9.98e0 | 2.02e1 | 4.16e1 | 8.63e1 | 1.79e2 | 3.72e2 | 7.75e2 |
| 64 | 1.52e1 | 3.12e1 | 6.45e1 | 1.34e2 | 2.80e2 | 5.84e2 | 1.22e3 |
| 128 | 2.34e1 | 4.82e1 | 1.00e2 | 2.09e2 | 4.37e2 | 9.14e2 | 1.91e3 |

**Table 2:** The eigenvector condition number $\kappa_{\text{eig}}(\tilde{\mathbf{A}}_H)$ when the optimization problem eq. (11) is solved with different values of $n$ and $\gamma$.

| $n$ \ $\gamma$ | 10 | $10^2$ | $10^3$ | $10^4$ | $10^5$ | $10^6$ | $10^7$ |
|---|---|---|---|---|---|---|---|
| 8 | 2.81e0 | 1.16e0 | 4.78e-1 | 1.98e-1 | 8.24e-2 | 3.45e-2 | 1.45e-2 |
| 16 | 6.77e0 | 2.86e0 | 1.22e0 | 5.18e-1 | 2.22e-1 | 9.50e-2 | 4.09e-2 |
| 32 | 1.62e1 | 6.96e0 | 3.00e0 | 1.30e0 | 5.62e-1 | 2.45e-1 | 1.07e-1 |
| 64 | 3.89e1 | 1.68e1 | 7.32e0 | 3.19e0 | 1.39e0 | 6.11e-1 | 2.69e-1 |
| 128 | 9.37e1 | 4.07e1 | 1.78e1 | 7.80e0 | 3.42e0 | 1.51e0 | 6.65e-1 |

**Table 3:** The perturbation size $\|\mathbf{E}\|$ when the optimization problem eq. (11) is solved with different values of $n$ and $\gamma$.

| Task | Depth | #Features | Norm | Prenorm | DO | LR | BS | Epochs | WD | $\Delta$ Range |
|---|---|---|---|---|---|---|---|---|---|---|
| ListOps | 8 | 256 | BN | False | 0. | 0.002 | 50 | 80 | 0.05 | (1e-3,1e0) |
| Text | 6 | 256 | BN | True | 0. | 0.01 | 16 | 80 | 0.05 | (1e-3,1e-1) |
| Retrieval | 6 | 128 | BN | True | 0. | 0.004 | 64 | 40 | 0.03 | (1e-3,1e-1) |
| Image | 6 | 128 | LN | False | 0.1 | 0.01 | 128 | 2000 | 0.01 | (1e-3,1e-1) |
| Pathfinder | 6 | 512 | BN | True | 0. | 0.004 | 64 | 300 | 0.03 | (1e-2,1e0) |
| Path-X | 6 | 128 | BN | True | 0. | 0.001 | 20 | 100 | 0.03 | (1e-4,1e-1) |
| Speech | 6 | 128 | BN | True | 0. | 0.01 | 16 | 40 | 0.05 | (1e-3,1e-1) |

**Table 4:** Configurations of the S4-PTD model, where DO, LR, BS, and WD stand for dropout rate, learning rate, batch size, and weight decay, respectively.

## J.1 DETAILS OF THE EVALUATION OF OUR MODEL IN THE LONG RANGE ARENA

To compare our perturbed models with the diagonal S4D and S5 models, we adopt the same model parameters used in Gu et al. (2022a) and Smith et al. (2023) but possibly change the training parameters, such as the learning rate, number of epochs, batch size, and weight decay rate. For choosing the perturbation matrix, we again solve the optimization problem in eq. (11). Instead of allowing $\gamma$ to be an unbounded positive tuning parameter, we require that $\gamma$ is large enough so that $\|\mathbf{E}\|/\|\mathbf{A}_H\| \leq 0.1$. This improves the worst-case robustness of our model (see section 5.2). We provide the detailed configuration of our S4-PTD model in Table 4 and that of our S5-PTD model in Table 5. In particular, we note that the first two columns of Table 4 are almost the same as those in Gu et al. (2022a)[9] and the first four columns of Table 5 match those in Smith et al. (2023) — these are model parameters. The only remaining non-trivial thing is that in the Path-X task, we start with a batch size of 32. We half the batch size after epoch 30 and epoch 60. By making the batch size smaller, we improve the generalization power of our model.

| Task | Depth | H | P | J | DO | LR | SSM LR | BS | Epochs | WD |
|---|---|---|---|---|---|---|---|---|---|---|
| ListOps | 8 | 128 | 16 | 8 | 0. | 0.003 | 0.001 | 50 | 35 | 0.05 |
| Text | 6 | 256 | 192 | 12 | 0.1 | 0.004 | 0.001 | 50 | 40 | 0.07 |
| Retrieval | 6 | 128 | 256 | 16 | 0. | 0.002 | 0.001 | 32 | 20 | 0.05 |
| Image | 6 | 512 | 384 | 3 | 0.1 | 0.0055 | 0.001 | 50 | 250 | 0.07 |
| Pathfinder | 6 | 192 | 256 | 8 | 0.05 | 0.0045 | 0.0009 | 64 | 230 | 0.05 |
| Path-X | 6 | 128 | 256 | 16 | 0. | 0.0018 | 0.0006 | 32 | 90 | 0.06 |
| Speech | 6 | 96 | 128 | 16 | 0.1 | 0.008 | 0.002 | 16 | 40 | 0.04 |

**Table 5:** Configurations of the S5-PTD model. See Smith et al. (2023) for the meaning of the parameter labels.

---

[9]The only exception is that in the Path-X task, we half the number of features in order to reduce the computation time. This only simplifies our perturbed model.

| Model | 16kHz | 8kHz |
|---|---|---|
| InceptionNet (Nonaka & Seita, 2021) | 61.24 | 05.18 |
| ResNet-1 (Nonaka & Seita, 2021) | 77.86 | 08.74 |
| XResNet-50 (Nonaka & Seita, 2021) | 83.01 | 07.72 |
| ConvNet (Nonaka & Seita, 2021) | 95.51 | 07.26 |
| S4 (Gu et al., 2022b) | 96.08 | 91.32 |
| Liquid-S4 (Hasani et al., 2023) | 96.78 | 90.00 |
| S4D (Gu et al., 2022a) | 95.83 | 91.08 |
| S4-PTD (ours) | 96.04 | 91.53 |
| S5 (Smith et al., 2023) | 96.52 | **94.53** |
| S5-PTD (ours) | **96.87** | 94.49 |

**Table 6:** Test accuracies on Speech Commands classification. We use the boldface number to indicate the highest test accuracy among all models for each task. We use the underlined number to indicate the highest test accuracy within the comparable group.

## J.2 DETAILS OF THE ROBUSTNESS TEST OF THE DIAGONAL MODEL AND OUR MODEL

In the robustness test presented in section 5.2, we train both an S4D model and an S4-PTD model. Our models have 4 layers, 128 channels, and each layer contains an SSM with $n = 32$ states. The perturbation matrix in the S4-PTD model is computed by setting $\gamma = 0.03$ in eq. (11). From Figure 3c, it can be seen that the perturbation thence computed has a magnitude of roughly $10\%$ of the magnitude of $\mathbf{A}_H$. We fix a universal discretization step $\Delta t$ (see Appendix B) for all channels. We leave the training dataset and the validation dataset unchanged, but we add $10\%$ of noises in the form of $\cos(325.4t)$ to the test dataset. The frequency 325.4 is chosen at one of the sensitivity regions of the diagonal SSM when $n = 32$. We train both models for 50 epochs and report the evolution of the training accuracy, the test accuracy on uncontaminated data, and that on noisy data. As mentioned in section 5.2, the matrix $\mathbf{A}$ is not trained (but see Appendix K for an example where $\mathbf{A}$ is trained).

## J.3 DETAILS OF THE ABLATION STUDY OF OUR MODEL

In section 5.3, we train models with different perturbation sizes to solve the sCIFAR task (Krizhevsky et al., 2009). Our models have the same architecture as those in the sensitivity test (see section 5.2). We set the batch size to be 64 and the learning rate to be 0.001 for SSM parameters and 0.01 for other model parameters. These are common setups that are adapted from the original S4 and S4D papers. We use the parameter $\gamma$ in eq. (11) to control the size of the perturbation $\|\mathbf{E}\|$. We set $\gamma = 10^{-4}, 10^{-3}, \ldots, 10^9$ and train the S4-PTD model for 200 epochs to learn a classifier. These correspond to the first 14 points in Figure 3c, where we report both the test accuracy of the model and the eigenvector condition number at initialization. Since setting $\gamma$ small does not help reducing $\kappa_{\text{eig}}(\tilde{\mathbf{A}}_H)$ all the way down to 1, the smallest condition number possible, to obtain the rightmost point, we perturb $\mathbf{A}_H$ by a random symmetric matrix with a large norm.

## K SUPPLEMENTARY RESULTS OF THE EXPERIMENTS

### K.1 FULL RESULTS ON ADDITIONAL DATASETS

In addition to the LRA dataset, we test our S4-PTD and S5-PTD models on the 35-way Speech Commands classification task (Warden, 2018), asking the model to identify a word in a recording from a pool of 35 words. The quantitative results are shown in Table 6.

### K.2 SUPPLEMENTARY RESULTS OF THE ROBUSTNESS TEST

In the robustness test presented in section 5.2, the state matrix $\mathbf{A}$ is fixed throughout training, and the non-robustness of the S4D initialization is subsequently observed at a particular frequency. In this section, we present supplementary results that aim to clarify the following two points:

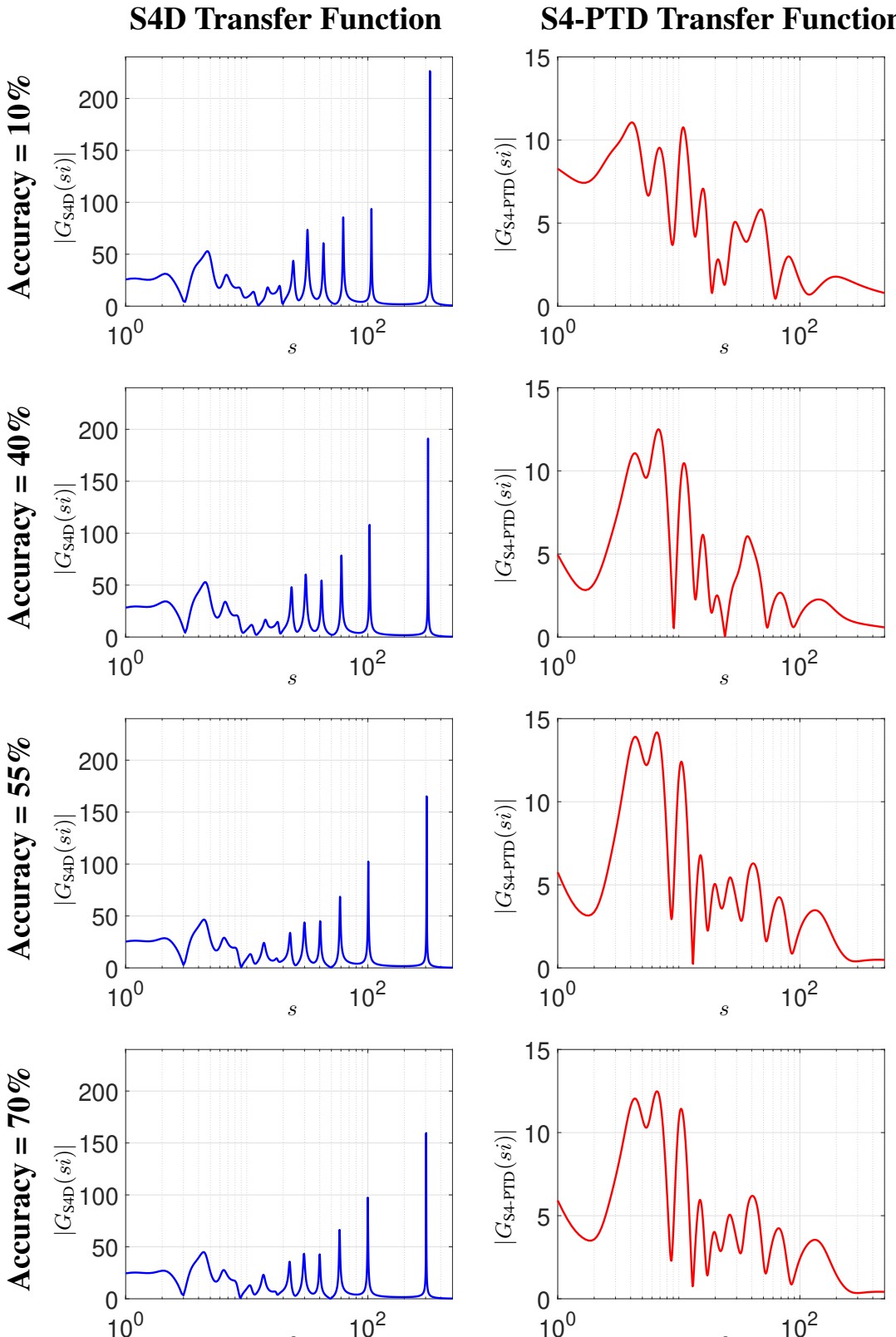

**Figure 13:** The evolution of the transfer functions in the S4D model (left) and the S4-PTD model (right). We observe that the spikes in the S4D model are preserved during the training while the S4-PTD model never develops any spike.

1. As discussed in section 5.2, the perturbation we used is the "worst-case" perturbation that makes S4D fail. In Figure 4, however, we showed that not all Fourier-mode perturbations have an essential effect. In fact, we revealed a strong correlation between the effect of the perturbation at a particular frequency and the location of the spikes in the transfer function of the S4D initialization. We provide more details of this experiment in this section.

2. While the state matrix $\mathbf{A}$ is not trained in the robustness test in section 5.2, we present a different experiment where we train the matrix $\mathbf{A}$ as usual. We then observe the evolution of the transfer function. We will empirically see that training does not solve the robustness issue of the S4D model.

In the first experiment (see Figure 4), we use the same setting as the one presented in Appendix J.2 but we contaminate the test set with a varying frequency $s$. From Figure 4(a), we observe a strong correlation between the effect of the perturbation and the location of the spikes in the graph of $G_{\mathrm{Diag}}$. In particular, a frequency $s$ located at such a spike drives the test accuracy to below $30\%$. This corroborates our discussion that the spikes in the graph of $G_{\mathrm{Diag}}$ impairs the worst-case robustness of the S4D initialization. We also show the analogous results for the S4-PTD model in Figure 4(b), where the size of the perturbation is chosen to be the same as the one used for LRA tasks (see appendix J.1). In this case, the performance of the S4-PTD model is relatively stable across all different modes of perturbation because the transfer function $G_{\mathrm{Pert}}$ is relatively smooth.

Next, we empirically study the behavior of the LTI systems during training. To do so, we again train a 4-layer SSM to learn the sCIFAR task, where all layers are either S4D layers or S4-PTD layers. Instead of fixing the state matrix $\mathbf{A}$, we also train this matrix. In particular, we set the learning rate of the state matrix $\mathbf{A}$ to be $0.004$ and the learning rate of the other model parameters to be $0.01$. We also apply a weight decay rate of $0.05$. This is standard practice in training an SSM. We then watch the evolution of the transfer functions of the LTI systems as the test accuracy increases. In Figure 13, we plot the evolution of both models as the test accuracy increases from $10\%$ (i.e., random guessing) to $70\%$. While there are many copies of LTI systems in a 4-layer, multi-channel SSM, we randomly select one and show its transfer function. We remark that other randomly selected LTI systems all show similar behaviors. From Figure 13, we see that while the heights of the spikes in the transfer function of the S4D LTI system decay slightly throughout training, they never vanish. In fact, the rate of decay gets smaller throughout training. The locations of the spikes also change, albeit by a small amount. Combining Figure 13 with Figure 4, we can conclude that training does not completely fix the robustness issue of the S4D model. In comparison, the transfer function of the S4-PTD model never develops any spike during the training process.

