# OpenReview forum: "Robustifying State-space Models for Long Sequences via Approximate Diagonalization"
_ICLR.cc/2024/Conference — ICLR 2024 spotlight_

### Official Review · Reviewer_X2Rr · 2023-10-30

**Soundness:** 3 good
**Presentation:** 3 good
**Contribution:** 3 good
**Rating:** 6
**Confidence:** 3

**Summary:**

Explores the diagonal state matrix used by recent linear SSMs such as S4D and S5 which were proposed as an approximation to the original diagonal plus low rank conjugation of the HiPPO matrix explored in the S4 paper. The work analyzes the differences between the S4 and S4D initialization through the transfer function and finds the convergence of S4D to S4 is not uniform. Experiments show that S4D models are sensitive to input perturbations which degrades the model's robustness. A proposed fix is adding a perturbation to the HiPPO matrix to allow for stable diagonalization. The proposed initialization is evaluated on the LRA tasks.

**Strengths:**

- The theoretical analysis seems sound and leads to several interesting insights including implications on the robustness of the initialization of the diagonal models.

- The empirical robustness experiments (synthetic and sCIFAR) help to illustrate the implications of the theoretical divergence

- The proposed fix is relatively straightforward and simple to implement, yet seems to lead to improved robustness results in the synthetic and sCIFAR tasks

**Weaknesses:**

- I am not convinced by the claim the S4-PTD model outperforms the S4D models on LRA. The LRU paper (https://arxiv.org/abs/2303.06349) reports results for S4D that are much better than the original reported S4D paper results. In addition, the appendix of the current paper under review states that mild hyperparameter tuning was performed for the S4-PTD models. Would mild hyperparameter tuning improve the S4D results as well?
  - To be clear, I am not claiming that S4-PTD must outperform S4D on all tasks to be a valuable contribution, simply that I think the claim about improving performance is too strong, and I would expect to see more results in a fair comparison to support this claim

- More experiment results on a broader range of robustness tasks would help strengthen the paper. At a minimum, perhaps modified versions of the speech and BIDMC tasks from previous papers in this line of work could have allowed for further evaluation of this method on some form of real world data.

- There seems to be a missing piece related to the training dynamics that I think would strengthen the paper. The theory is related to the dynamics matrix $A$ at initialization, but the $A$ matrices are trained during the experiments. How do the trajectories of the eigenvalues differ during training such that the PTD models are more robust during the perturbation experiments? This should be relatively to simple to track during training.

- While it is true the proposed perturbation method could be applied to any dynamics matrix, no other matrices are explored. Perhaps some of the other diagonal matrices proposed in the S4D paper could have been experimented with. Would a similar approach be beneficial if applied to initialize the effective dynamics matrix of the Liquid-S4 model?

- The paper doesn't really seem to support the claim that the S5 models suffer from the same perturbation
issue. I believe this is likely the case, but no empirical results are presented that support this. Can an example of S5 and S5-PTD be included in the perturbation experiments also?

**Questions:**

Please see weaknesses above.

---

> ### Author Response · Authors · 2023-11-18
> **Response to the reviewer**
>
> We thank the reviewer for the careful review and insightful comments. Below we address the questions and comments raised in this review.
>
> > **Weakness 1** "I am not convinced by the claim..."
>
> **Response:** We thank the reviewer for pointing the new results out. First, we clarify that by saying "mild tuning," we do not mean a laborious grid search. Instead, what we really want to say is that we did not take the exact hyperparameters from S4D/S5 because the models are initialized differently so the S4-PTD/S5-PTD models are expected to have a different set of good hyperparameters. The S4D/S5 models must have also been tuned and we did not encounter a better set of hyperparameters that can significantly improve the S4D/S5 performance.
>
> The LRU paper has performed a more careful tuning than what we have done. In addition, the result reported for the Image task is based on colorful sCIFAR instead of the grayscaled one as used in the original S4D/S5 paper and ours. In the appendix of the LRU paper, the authors also report some architectural differences, e.g., they used a different activation function for the PathX task and it seems that the number of model parameters is also different from what is used in the original S4D/S5 paper and our manuscript. We will reach out to the authors of the LRU paper to get more information on the details of the S4D model they have tuned. We will make sure that we have an unbiased claim about the performance of our models in the camera-ready version should the paper be accepted.
>
> > **Weakness 2** "More experiment results on a broader range..."
>
> **Response:** We have done some experiments on the Speech Command task (see Table 6). We are working on the BIDMC task and will be sure to include the result in the camera-ready version should the manuscript be accepted. We remark that due to the limited time, we could not tune the models for Speech Command. The hyperparameters are exactly copied from the S4D/S5 models. Hence, the results reported may not be optimal.
>
> > **Weakness 3** "There seems to be a missing piece..."
>
> **Response:** We have added an empirical evaluation of the evolution of the transfer function over training (see Figure 13). We find that the instabilities in the S4D model are preserved throughout training while the S4-PTD model does not develop any instability. We hope this adds a new perspective to our paper and makes it more complete.
>
> > **Weakness 4** "While it is true the proposed perturbation method..."
>
> **Response:** We are interested in comparing the S4-PTD/S5-PTD models with their S4D/S5 companion initialized by other matrices. However, since we cannot find the full results on large S4D/S5 models, there lacks a benchmark to be compared with. We are happy to reproduce the S4D/S5 results with other initialization schemes and will include a discussion about it should the paper be accepted.
>
> As for Liquid-S4, the proposed perturbation method can be applied. The original Liquid-S4 model leverages the DPLR structure of $\mathbf{A}$, which corresponds to the S4 model. However, we believe if one uses a diagonal $\mathbf{A}$, then the perturbed initialization makes the model more robust than as if the model is initialized by the S4D diagonal initialization. Since our method only changes the initialization, it is simple to incorporate it into a diagonal variant of the Liquid-S4 model.
>
> > **Weakness 5** "The paper doesn't really seem to support..."
>
> **Response:** It is very likely that the S5 initialization suffers from the same issue, as suggested by Figure 4. We have implemented an experiment to test the robustness of the S5/S5-PTD models. We will update our manuscript when the results come in.
>
> We hope this answers the reviewer's questions and concerns. We are happy to answer any follow-up question(s) that the reviewer may have.

---

> > ### Comment · Reviewer_X2Rr · 2023-11-21
> >
> > Thank you for the clarifications. I have increased my score.

---

> > > ### Author Response · Authors · 2023-11-21
> > > **Thank you for your reply.**
> > >
> > > We thank the reviewer for going through the rebuttal and re-evaluating our manuscript.

---

### Official Review · Reviewer_faKE · 2023-10-30

**Soundness:** 2 fair
**Presentation:** 3 good
**Contribution:** 3 good
**Rating:** 6
**Confidence:** 3

**Summary:**

This paper presents a theoretical and empirical study of the frequency-domain implications of various approximations common across deep state space models.  The authors make the connection between the structure of approximation, non-normality of operators, and the non-uniformity of the approximation across frequencies.  This pathology is tested across several synthetic examples. This insight is used to propose a new initialization scheme, which appears to ameliorate the degeneracy.

**Strengths:**

The paper itself is pretty well written, with figures and tables prepared excellently.  I believe the analysis is both a novel and useful insight into this new class of model. The theoretical analysis itself appears correct (although I did not go through the proofs in detail).  The experiments conducted also back-up the theoretical claims (modulo my comments below).

**Weaknesses:**

I think the paper is pretty sound.  My main concerns are about some of the presentation choices, and the “secret sauce” in the contribution.  I have tried to outline some concrete weaknesses here that, if remedied, would improve the paper.  I have then tried to write out my slightly more high-level question in Questions.

**W.1.: The core method**:  Please can the authors clarify what the method actually entails.  As far as i can tell, the “method” essentially specifies an initialization for S4D, given in (10) followed by diagonalization.  This is used instead of the diagonalization of the Normal component of the HiPPO matrix in S4D.  If this is true, this should be absolutely, concretely clearly stated somewhere.  I think right now it is tied up in the introduction with spectral analysis, PTD, backwards/forwards approximations etc.

**W.2.: Choice of what the paper emphasizes**:  The authors have clearly done _a lot_ of theoretical analysis.  However, I think the net result is an incredibly simple result – adding a random perturbation makes models more robust.  I really think this result, and any usable empirical analysis of this result, should be highlighted.  Some of the theoretical points should be relegated to the appendix.  If the results are to be believed, then the banner result should be “ignore DPLR/diagonal state spaces and use random perturbations”.  This core message is not highlighted.

Similarly, there are some nice experiments in the supplement demonstrating the pathology experimentally, e.g. fig 7-10.  These should be brought up to the main, as they help flesh out the arguments far more than a detailed introduction to the intricacies of the conjugation of HiPPO or some of the theorems.  I think this will help improve the reach and impact of the paper.

**W.3.: Figure 2 results**:  I find the results in Figure 2 very hard to parse out.  The core point (to my understanding) is that S4-PTD is a faithful approximation of S4.  However, I do not necessarily see why S4Ds predictions _should_ be reminiscent of S4 – it’s a different model! – or that it should be worse in the way predicted.  I think it is coincidental that S4D is so much worse than S4(-PTD) on this task.  I ask the authors to clarify the intuition and results of this experiment.  (See also Q.1.)

I think it is also somewhat coincidental that S4-PTDs predictions look like S4 – surely different random seeds should result in different predictions?

**W.4.: Missing definitions and exposition**: Lots of terms are basically undefined, e.g.:
- The introduction of “backward stability” is deferred to the appendix, but is really the core of the method.  There is no discussion in the main really of why forward/backwards even relevant, or why existing methods fall down here.
- Pseudospectral theory is undefined, nor why adding a random perturbation ($E$) helps.
- Why spectral information with a poor condition number is “meaningless”.

This makes the contribution a little less self-contained.  I would rather drop some of the theoretical analysis from the main to the supplement and bring up more of the core definitions.  This would dramatically improve the impact of the work.

**W.5.: No conclusion or discussion**:  Not really a weakness, but there is no conclusion, discussion or outlined future work.  It would be nice to see some critical self-reflection on the work.  It would also help you reinforce the core messages you are trying to convey in the paper.

## Minor weaknesses
- (e.g.) “section 3.4” should be capitalized.
- The explanation of Figure 2 on Page 6 is very poor.
- use \citet and \citep where appropriate.
- Author style in the bibliography is inconsistent.

**Questions:**

**Q.1.**  This is a very general question, but something that I cannot wrap my head around.  I invite the authors to try and help me understand.  (The remedy to this will also affect my perception of the weaknesses)

Throughout the text this is an analysis of (e.g.) HiPPO initialization, but the results presented are for trained models.  I am surprised that the pathology predicted by the initialization is preserved through training quite so neatly.  For instance, in Figure 3, how do you obtain the frequencies that the model should be perturbed at?  Do you inspect the trained model, or the a-priori specified HiPPO matrix?  Each channel in the S4D model should have different frequencies, and so are you corrupting them all?  Or just one?

My follow-up is why does the approximate diagonalization on initialization lead to more robust trained models?  Presumably $E$ isn’t learned?  Once the model is diagonalized, the initialization is essentially thrown away.  As a result, there is a real disconnect in Figure 3.  It looks like S4-PTD might be _slightly_ worse than S4, but the training curves are ostensibly identical.  But the robustness to perturbations is clear.  Why do these pathological frequencies survive training in S4?

I think there might be two contributions here that are currently wrapped up in one-another.  The first contribution is an analysis of the frequency properties of a hypothetical (single) SSM.  The second contribution is an initialization scheme for systems of deep SSMs that is derived from some of this insight, but doesn’t necessarily directly solve the initial problem.

For instance, I would like to see an experiment sweeping over the perturbation frequency (e.g. combining Figures 1 & 3), and showing that there are certain choice frequencies that destroy performance, as opposed to just a model that is less robust to any perturbations.  I think this is an important distinction that is somewhat predicted by the theory, but is unverified.  I think this is what Figure 6 might be driving at, but to my eye, it looks like the diagonal system is just unilaterally worse than DPLR.  This would also be an empirical validation of Figure 1.  I would like to see the results for an untrained system, a single trained DPLR system, and also for an S4D model (where there are multiple S4 systems in parallel, each with their own characteristics).

In that vein, I'd also like to see this graph traced out over training and across different input frequencies.  E.g. If the training data has a lot of power in a frequency where there is a peak in the transfer function, then how does/does the model learn to move that peak away from that frequency?

I invite the authors to comment on this.  I realize my thoughts on this are a little jumbled, but I think this is maybe emblematic of a shortcoming in the presentation or cross-linking of the ideas.  If the authors want more clarification, I will happily try and restate my concerns.

**Q.2:**  Why is it relevant that the final peaks in Figure 1 are all (roughly) the same height?

---

> ### Author Response · Authors · 2023-11-18
> **Response to the reviewer (1/2)**
>
> We thank the reviewer for the careful review and insightful comments. Below we address the questions and comments raised in this review.
>
> > **Weakness 1**  "The core method"
>
> **Response:** Yes, this is indeed a main practical contribution of the paper. We have highlighted this in several places:
> * In the last paragraph of the introduction (right above the contributions), we have explained the S4-PTD and S5-PTD models in a succinct way and pointed out their advantage in terms of robustness.
>
> * We have also highlighted this method in Contribution \#4.
>
> * In section 4, we explained the model concretely in (9) and the text around it. In addition, at the end of the first paragraph, we stated how the S4-PTD and S5-PTD models are different from the S4D and S5 models.
>
> * Finally, we also highlighted our contribution in our conclusion.
>
>
> > **Weakness 2**  "Choice of what the paper emphasizes"
>
> **Response:** We agree that more empirical analysis of our theory and model can be informative. In addition to highlighting the main practical message of the paper, as explained above, we have made the following changes:
>
> * We have put Lemma 1 and its discussion into the appendices. In addition, we have also moved the experiment on the interpolation/extrapolation into the appendix; instead, we now use Figure 4 to more efficiently deliver the same message.
>
> * We have added Figure 2 (previously as Figure 8-9), in which the diagonal and the DPLR systems are simulated given both a smooth input and a non-smooth input. This helps the readers to visualize and better understand Theorem 1.
>
> * We have also added Figure 4, which is a new experiment added to the manuscript. The main message is that we have verified the spikes in the transfer function of the S4D initialization impair its robustness. This is also related to our answer to Q1 raised by the reviewer (see below).
>
> > **Weakness 3**  "Figure 2 results"
>
> **Response:** First, we have put this experiment into the appendix because we believe our new result (see Figure 4) helps better explain our main message. It is not coincidental that the S4D model is much worse than the S4 and S4-PTD models in this case. This can be understood from Figure 4. The main reason is that the S4D model is not stable near certain Fourier modes. Hence, it is hopeless to do interpolation or extrapolation near those frequencies if they are not included in the training dataset. We believe the fact that the S4 model and the S4-PTD model behave similarly is not merely coincidental either. If one looks into the low-frequency domains in Figure 9 (previously as Figure 2), then the S4 and the S4-PTD models behave very differently. We believe the main reason that the two models behave similarly in the high-frequency domain is that we observe, empirically, that most changes in the LTI systems during training only happen in the low-frequency domain. (See Figure 13 and our answer to Q1.) We think this explains why the two models predict similarly in the high-frequency domains but not the low-frequency ones.
>
> > **Weakness 4**  "Missing definitions and exposition"
>
> **Response:** We have added some definition/explanation to clarify these terminologies/statements:
>
> * We have added a paragraph in section 4 (see the second paragraph that starts with "Consider the process of diagonalizing...") to define the meaning of the backward and forward errors in the context of SSMs. In the same paragraph, we have also explained from a high-level perspective why it suffices to consider the backward error and not the forward error. We believe this is a good place to introduce the two types of errors in detail because they are not used until Theorem 3.
>
> * We have added a footnote (see footnote 2) to explain what the pseudospectral theory is mainly about. The pseudospectral theory studies a very broad class of questions and is not just a single theorem. For example, Theorem 4 about the eigenvector condition number under perturbation is a result that falls under the pseudospectral theory and explains why adding the perturbation matrix $\mathbf{E}$ helps.
>
> * The true spectral information of a highly non-normal matrix is meaningless because it is very unstable. Hence, using the spectral information, one would suffer from a very large variance (in terms of the bias-variance tradeoff). The other way to view it is that when $\mathbf{A}$ is highly non-normal, the eigenvector matrix $\mathbf{V}$ is very ill-conditioned. Hence, for example, one cannot project a vector $\mathbf{v}$ onto the eigenbasis and use the spectral information of $\mathbf{A}$ to study $f(\mathbf{A})\mathbf{v}$, where $f$ is an analytic function. We have added a brief version of this discussion as a footnote (see footnote 1).
>
> > **Weakness 5**  "No conclusion or discussion"
>
> **Response:** We have added a conclusion to summarize the manuscript and suggest future research avenues.

---

> ### Author Response · Authors · 2023-11-18
> **Response to the reviewer (2/2)**
>
> > **Weakness 6**  "Minor weaknesses"
>
> **Response:** We have made the following changes:
>
> * In the ICLR style guideline, sections are not capitalized. We follow this convention in our manuscript.
>
> * We have improved our explanation in the caption of the figure. Note that we have moved the figure into the appendix (see Figure 9).
>
> * We have corrected all misuses of \citet and \citep.
>
> * We have made the names of the authors consistent in our bibliography.
>
> > **Question 1**  "This is a very general question, but..."
>
> **Response:** We thank the reviewer for this very insightful question. This question has inspired some discussion that we did not include in our manuscript initially. We have added a more detailed explanation of the experiment setup in the main text and Appendix J.2 and a fair amount of empirical discussion (see Appendix K.2) which we believe will clarify and strengthen the paper. Hopefully, the following points we make (partially) answer the reviewer's question:
>
> * In the experiment shown in Figure 3(a/b), we fixed the state matrix $\mathbf{A}$. The main reason for fixing the matrix is that, as explained later on, training slightly shifts the locations of the spikes in the transfer function of the S4D LTI systems, which requires us to perform a case study after each epoch to find a frequency $s$ at which the S4D model is not stable. We have made this experimental setup clear in our manuscript to avoid confusion.
>
> * We have added a new appendix (See Appendix K). In Figure 4 and Appendix K.2, we show an experiment where we contaminate the test dataset with Fourier noises at varying frequencies $s$. This is still done with a fixed state matrix $\mathbf{A}$. In Figure 4, we plot the transfer function of the S4D initialization on top of the test accuracy, which reveals a strong correlation between the "spikes" in the transfer function and the effect of the perturbation at the corresponding frequencies. This connects our theoretical study of the S4D transfer functions to the discussion of the robustness of the model.
>
> * Now, we consider the case where we indeed train the state matrix $\mathbf{A}$. In this new experiment, we train our SSMs as everyone does in practice (see Appendix K.2 for more details). Given our observation in Figure 4, we use the transfer function as a protocol to evaluate the robustness of our model. We watch the evolution of the transfer function (randomly selected from many transfer functions in the SSM) throughout training. The results are shown in Figure 13 and here is a summary of what is going on:
>
>    * The spikes in the transfer function of the S4D model have slightly decayed but never vanished. We believe this slight decay is possibly due to the weight decay rate we set for the LTI matrices.
>
>    * The locations of the spikes shift very slightly. This is the main reason why when conducting the experiment in Figure 3(a/b), we fixed the matrix $\mathbf{A}$: otherwise, we would have to eyeball the location of the spike after each epoch.
>
>    * The transfer function of the S4-PTD model never develops any spike or instability throughout training.
>
> In conclusion, this shows training does not completely solve the robustness issue of the S4D model. Neither does it impair the robustness of the S4-PTD model.
>
> Since this paper is mainly about analyzing the initialization, we chose to put all these discussions into the appendices and leave them as a remark in the main manuscript. Understanding the training dynamics of the LTI systems is certainly an interesting future research avenue. We do remark that in our conclusion. We are happy to answer any follow-up questions that the reviewer may have and also include a more thorough empirical evaluation along this line in our camera-ready version should this manuscript be accepted.
>
> > **Question 2**  "Why is it relevant that the final peaks..."
>
> **Response:** The fact that the peaks are all roughly the same height means that $G_{\text{Diag}}$ does not converge to $G_{\text{DPLR}}$ uniformly. Hence, given any fixed state space size $n$, there is always a Fourier mode $\mathbf{u}$ on which the diagonal system and the DPLR system produce very different results. In other words, if the peaks vanish as $n \rightarrow \infty$, then by taking $n$ large enough, the transfer function $G_{\text{Diag}}$ will become a smooth function so it is robust to any Fourier-mode perturbation. However, due to the non-vanishing peaks, the S4D initialization is always not robust to _some_ Fourier-mode perturbation no matter how large $n$ is. This can be seen from Figure 4.
>
> We hope this answers the reviewer's questions and concerns. We are happy to answer any follow-up question(s) that the reviewer may have.

---

> ### Comment · Reviewer_faKE · 2023-11-20
> **One step forward; one step back.**
>
> To the authors,
>
> Thank you for your incredibly detailed response and for updating the paper.  I think the additions and clarifications have definitely strengthened the paper.
>
> However, I think I will keep my score at a six.  If there were a seven, I would upgrade to that, but the jump to eight is too large.  The amount the paper changed, inspired by feedback from all four reviewers, suggests that the paper was not quite ready for submission.  I think the clarity has taken a backward step during the review process, even if the science has moved forward.  There is a lot of great work in there, unquestionably;  but the ideas are a bit jumbled, and there is a lot of work to be done to iron out the flow and linking of ideas to present a coherent, unified, usable message.
>
> I believe there is an excellent conference submission in this work, I just don't believe it is quite there yet.
>
> Thanks, and good luck,
>
> faKE*
>
> (*LOL)

---

> > ### Author Response · Authors · 2023-11-21
> > **Thank you for your reply.**
> >
> > We appreciate the reviewer's positive feedback, “raising” the score to an imaginary "7." We are seeking further clarification to potentially reach an "8" within the remaining time. We strongly feel that our revisions address initial concerns and enhance the paper's quality. We made changes to directly respond to reviewers' comments, aiming to improve clarity without obscuring core ideas. Specifically, we
> >
> > * replaced technical content in section 3 with a numerical experiment,
> > * simplified the interpolation/extrapolation experiment into the self-evident Figure 4,
> > * added highlights in the introduction and conclusion, and
> > * added a self-contained discussion of the backward/forward errors.
> >
> > Aside from these, no major revisions affecting clarity were made. It would be helpful for us if the reviewer could kindly point to specific clarity issues so that we can address them.

---

### Official Review · Reviewer_eMiw · 2023-10-31

**Soundness:** 4 excellent
**Presentation:** 4 excellent
**Contribution:** 3 good
**Rating:** 8
**Confidence:** 5

**Summary:**

This paper presents a deep theoretical analysis of various aspects of S4 models.

First, it provides a much more fine-grained analysis of the difference between the S4-DPLR and S4-Diag models, the two most well-known instantiations of S4 models (more precisely with the "HIPPO LegS" initializations), than prior work, including
- Exact analysis of the difference in transfer functions
- This allows providing non-asymptotic analysis of the convergence between the DPLR HIPPO init and its diagonal approximation by dropping the rank-1 term. As opposed to prior work which only showed it asymptotically in the limit of state size $n$
- This also leads to better understanding of what type of input perturbations lead to divergence between the approximation

The technical machinery leads to a very simple but well-motivated solution to the stability issues of S4D, by perturbing the $\mathbf{A}$ matrix before diagonalizing. This method presents modest empirical improvements over the original methods.

**Strengths:**

1. Extremely clearly written technical background and presentation of technical results.
2. The discussion of the relationship of S4-DPLR and S4-Diag from a functional analysis and transfer function perspective is strong and substantially contributes to the theoretical understanding of these popular models. The theoretical derivation of their properties and validation through synthetic experiments is well-done.
3. The main motivation of the perturb-then-diagonalize (PTD) being justified because only backwards error (instead of forwards error) matters is a nice idea.
4. The empirical results are strong and ablations are well-motivated.

Overall, the paper has high originality, quality, and clarity.

**Weaknesses:**

The only potential weakness of this paper is the practical significance to the machine learning community. To my knowledge, in practice most applications of S4 use the diagonal variants to no noticeable detriment, and also often use the simpler S4D-Linear initialization to which the present theory has less practical implications for. Put another way, while technically strong, it is not clear whether the ideas can lead to truly new capabilities for machine learning models.

On the other hand, this paper does provide a new direction toward revitalizing interest in the HIPPO theory and other (perhaps yet undiscovered) flavors of SSMs, which may still provide practical benefits in more specific regimes (e.g. limited data or when a strong inductive bias towards the memorization interpretation of HIPPO is desired). And as a mainly theoretical paper with strong contributions towards the theory of these popular models, in the context of this paper this is not a major weakness.

**Questions:**

End of Sec 3.1:
> Moreover, for every $n \ge 1$, zooming into the last spike...

1. is the constant magnitude a conjecture, or easy to show from Lemma 1? How were the final spike locations derived? Seems like it should be possible to find the exact location for every $n$ and calculate the magnitude of the spikes from Lemma 1, although maybe the calculations are actually difficult.
2. I'm confused how to interpret left side of the plot in relation to the right: are the final yellow/orange/blue spikes supposed to actually go up to magnitude $\approx 1$ (same as the purple spike), but the sampling of the plot x-axis is too coarse grained to get the exact location of the spike so is not rendered precisely?
3. The text claims the last spike is located at $|s| = \Theta(n^2)$ but it seems like larger $n$ correspond to earlier spikes (this seems important as it is related to a later question)
4. Is there a limiting result of the nature of $G_{\text{Diag}}(n \to \infty)$? These plots seem to suggest that the width of the spikes also decreases, but the rate is not clear. Depending on the nature of this limit, it's not actually clear to me that $|G_{Diag}(si)|$ does not converge uniformly: the domain is bounded from below ($s \ge 0$) so the location of the spike can't move too fast; if the spikes also have width that does not decrease too fast, then it seems like it actually could converge uniformly? On the other hand if the domain is transformed logarithmically as in Figure 1, then I would believe that $G_{Diag}(e^s i)$ does not converge uniformly, so maybe that's what you mean? (Also, is there a reason Figure 1 only plots frequencies $\ge 1$?)

In Section 3.3:
> the outputs of the two systems diverge given the unit impulse (i.e., the Dirac delta function) as the input

5. I'm not sure of the accuracy of this statement, or at least am confused how it relates to results from the S4D paper. If I understand correctly, the outputs of the two systems given the unit impulse should just be the impulse response or the "continuous convolution kernel" $K(t) = C e^{tA} B$. However the S4D paper indicates that this does converge, at least pointwise. (See S4D Figure 2, and the corresponding reproduction: https://github.com/HazyResearch/state-spaces/blob/main/notebooks/ssm_kernels.ipynb) To my understanding those results are generated in the same setting as Appendix F.3, Figure 9 in this submission, but the results look quite different.
6. Additionally the associated discussion says "The oscillatory behavior can be explained by our observation in Figure 1: the larger the $n$, the later the spike emerges. This means that for a larger $n$, the outputs of two systems differ at a higher frequency (i.e., a more oscillatory mode)". But Figure 1 seems to say that larger $n$ corresponds to spikes at lower frequencies (which is related to another question above)

7. Section 3.4: I think the description of the experiment does not state which $n$ is used; based on Fig 1, it seems like it is fixed to $n=10000$?

8. Theorem 3: how tight is the upper bound empirically? This result says that as $n$ grows, even if the total norm of the error matrix is controlled (hence the entries decrease), the total output deviation still increases with larger $n$. In practice, does the output deviation increase with state size?

9. How is the hyperparameter $\gamma$ set for each dataset? It doesn't appear in the hyperparameter table. From Figure 3(c) it seems as if the results can be somewhat sensitive to this parameter.

Overall I think this paper is a very interesting and technical deep-dive into the theory of S4 models.
I am happy to increase my soundness score and overall score after discussion of some of these questions.

---

> ### Author Response · Authors · 2023-11-18
> **Response to the reviewer (1/2)**
>
> We thank the reviewer for the careful review and insightful comments. Below we address the questions and comments raised in this review.
>
> > **Question 1**  "Is the constant magnitude a conjecture..."
>
> **Response:** The constant magnitude of the last peak is a proved statement in our manuscript. It is proved in Appendix D: Proof of Theorem 2. (See the part around ``If we can show that $s_n=\Omega(n^2)$, then we have that $|G_{\text{DPLR}}(s_ni)-G_{\text{Diag}}(s_ni)| = \Omega(1)$...") In particular, we explicitly show that the height of the spikes stay constant and then use this fact to show that $G_{\text{DPLR}} -G_{\text{Diag}}$ does not converge to zero uniformly (i.e. Theorem 2). The calculation of the exact location of the last spike is difficult, involving solving a complicated trigonometric equation. However, we derived an asymptotic expression for it based on the angle of the denominator of the expression in Lemma 1 (see (24) in Appendix D).
>
> > **Question 2**  "I'm confused how to interpret left side of the plot..."
>
> **Response:** The reviewer is correct about the interpretation of the right panels. The big panel on the left does not have enough resolution to reveal the correct height of the spikes, and in the right panels, we zoom in to see that the height of the spike remains constant. We have added a clarification in the figure caption.
>
> > **Question 3**  "The text claims the last spike is located at..."
>
> **Response:** We apologize for the incorrect legend in Figure 1. The correct order from top to bottom should correspond to $n = 10000, 1000, 100$, and $10$, respectively. We thank the reviewer for pointing out this issue. We have corrected the legend.
>
> > **Question 4**  "Is there a limiting result of the nature of..."
>
> **Response:** Again, we believe this is another instance of confusion caused by the incorrect legend. With the corrected legend, we see that the spikes are moving rightward. Depending on the interpretation of the "width", it can be either increasing or staying constant. For example, if the "width" is defined visually as the length of the interval on which the graphs of $G_{\text{Diag}}$ and $G_{\text{DPLR}}$ visibly differ, then it increases as $n \rightarrow \infty$. However, fixing a small constant $\delta > 0$, if one defines the "width" of the spike as the measure of the interval for which $|G_{\text{DPLR}}| > \delta$, then we proved in Appendix E that the width stays constant as $n \rightarrow \infty$. This can also be verified by the right panels in Figure 1. The difference here is that in the first definition, $\delta = \delta(n)$ depends on $n$ and decreases as $n \rightarrow \infty$, whereas in the second definition, we fix a $\delta$. We believe with the corrected legend, there is no confusion about the range of the $x$-axis. We are happy to answer any further questions that the reviewer has about Figure 1.

---

> ### Author Response · Authors · 2023-11-18
> **Response to the reviewer (2/2)**
>
> > **Question 5**  "I'm not sure of the accuracy of this statement..."
>
> **Response:** The reviewer made a very good point here. First, we confirm that the output from the Dirac delta is equivalent to the convolution kernel used in the S4D paper. The real difference here is that the S4D paper mainly concerns the discretized system while our theory considers the continuous-time systems. For example, in S4D Figure 2, there is a distinction between the ZOH and the bilinear discretizations. However, from a continuous perspective, this distinction should not exist.
>
> Now, we explain why there seems to be a dichotomy between our result and the S4D result. We are unable to post pictures on OpenReview, but we kindly advise the reviewer to run the following three commands in the Jupyter Notebook (https://github.com/HazyResearch/state-spaces/blob/main/notebooks/ssm_kernels.ipynb):
>
> ```
> plot(N=128, measure='legsd', T=2.5, dt=0.01, yticks=[-1.0, 0.0, 1.0, 2.0], disc='bilinear')
> plot(N=128, measure='legsd', T=2.5, dt=0.001, yticks=[-1.0, 0.0, 1.0, 2.0], disc='bilinear')
> plot(N=128, measure='legsd', T=2.5, dt=0.0001, yticks=[-1.0, 0.0, 1.0, 2.0], disc='bilinear')
> ```
>
> Then, the reviewer will notice that the convolution kernel of the S4D initialization under the discretization step $dt = 0.01$ is almost identical to that of the S4 initialization, whereas the convolution kernel under the discretization step $dt = 0.0001$ is wildly different.
>
> Why does this happen? The reason is the so-called aliasing error [Trefethen, 2019, Ch. 4]: under a coarse discretization, the high frequencies cannot be observed. For example, one cannot tell the difference between $\cos(10\pi t)$ and $\cos(1000\pi t)$ by samples at $t = k\pi$ ($k \in \mathbb{Z}$). The same story applies here. If a discretization step $dt$ is fixed a priori, then there is a threshold $S_{dt}$ such that any frequency $s > S_{dt}$ collapse into a frequency lower than $S_{dt}$. Hence, if $n$ is sufficiently large, then by our theory, the S4 initialization and the S4D initialization differ mostly in a region of high frequencies. If this region goes beyond $S_{dt}$, then the two convolution kernels will appear almost identical. However, since our analysis is on a continuous level, this pathology will not appear because our "discretization step" $dt$ is truly infinitesimal. In conclusion, from a continuous perspective, the impulse response of the S4D initialization never converges to that of the S4 initialization. However, convergence of the discretized convolution kernel happens if a discretization step $dt$ is fixed a priori. We have also left this as a remark in our main text. We hope this addresses the reviewer's concern.
>
> Lloyd N. Trefethen. Approximation Theory and Approximation Practice, Extended Edition. Society for Industrial and Applied Mathematics, Philadelphia, PA, 2019.
>
> > **Question 6**  "Additionally the associated discussion says..."
>
> **Response:** This is another confusion caused by the wrong legend in Figure 1. We apologize for that.
>
> > **Question 7**  "Section 3.4: I think the description..."
>
> **Response:** We fixed $n = 32$ in this experiment, which is the default state-space size in running many ablation studies and examples from previous SSMs works. We have added this detail to Appendix G (previously as section 3.4). Again, it is our mislabeled legend in Figure 1 that gave the reviewer the false impression that $n = 10000$.
>
> > **Question 8**  "Theorem 3: how tight is the upper bound..."
>
> **Response:** We have conducted an empirical simulation to test the perturbation of the transfer function. The results have been added to Appendix I.1 (See Figure 11). In summary, we observed that as $n$ increases, if we fix the norm of $\mathbf{E}$, then in the average case, the transfer function perturbation decays with respect to $n$. As the reviewer noted, in this theorem, we fixed the norm of $\mathbf{E}$, so it is not surprising that $\\|G_{\text{DPLR}} - G_{\text{Pert}}\\|_{\infty}$ decreases as $n \rightarrow \infty$ in the average case because the relative perturbation size $\\|\mathbf{E}\\|/\\|\mathbf{A}\\|$ also decreases. On the other hand, Theorem 3 is a worst-case error estimate and we have added a theoretical discussion about the dichotomy between the worst-case behavior and the average-case behavior in Appendix I.1 as well.
>
> > **Question 9**  "How is the hyperparameter..."
>
> **Response:** In Appendix J.1, we revealed that for the LRA dataset, we choose $\gamma$ so that the size of the perturbation is about 10% of the size of $\mathbf{A}_H$. By Figure 3(b), this still empirically guarantees that our initialization is robust under the worst-case Fourier mode perturbation. We adopt the same convention for the interpolation/extrapolation experiment (now moved into the appendices).
>
> We hope this answers the reviewer's questions and concerns. We are happy to answer any follow-up question(s) that the reviewer may have.

---

> ### Comment · Reviewer_eMiw · 2023-11-23
> **Post Author Response**
>
> Thank you for the additional clarifications. Indeed, many (perhaps most) of my original confusions were a result of the erroneous caption about the spike locations increasing or decreasing. My question about the convergence of S4D and S4 was also answered by the distinction between the continuous and discrete point of views (I think this discussion would be useful somewhere directly in the paper, at least in the corresponding Appendix). Overall, I think this is a strong theoretical paper and am increasing my score.

---

> > ### Author Response · Authors · 2023-11-23
> > **Thank you for your reply.**
> >
> > We thank the reviewer for going through the rebuttal and re-evaluating our manuscript. We will be sure to add a discussion of the distinction between the continuous and discrete kernels in our appendix in the camera-ready version should the paper be accepted.

---

### Official Review · Reviewer_Xrm4 · 2023-11-01

**Soundness:** 3 good
**Presentation:** 3 good
**Contribution:** 3 good
**Rating:** 6
**Confidence:** 2

**Summary:**

The authors study the State-space models (SSMs), which have shown promising results on long range sequence tasks. To speed up SSMs, diagonal approximation has been considered, which works well in practice but lacks theoretical guarantees. In this work, the authors propose a generic, backward stable perturb-then-diagonalize (PTD) strategy. The theoretical analysis shows convergence guarantees and proves the stronger robustness of the proposed method. The empirical evaluation also shows strong performance of the proposed method with the compared baselines.

**Strengths:**

1. The proposed method has theoretical guarantees for convergence and robustness.
2. The empirical result shows solid performance of the proposed method.

**Weaknesses:**

1. The writing is quite dense and can be improved if the authors make it more self-contained.

**Questions:**

1. Can the author elaborate more on why the proposed method can outperform the ones without diagonalization? This seems to be quite interesting.

---

> ### Author Response · Authors · 2023-11-18
> **Response to the reviewer**
>
> We thank the reviewer for the careful review and insightful comments. Below we address the questions and comments raised in this review.
>
> > **Weakness 1**  "The writing is quite dense and can be..."
>
> **Response:** We have improved the presentation of our paper. While a list of major changes we have made is summarized in our general comment above, we highlight some of them that particularly made the manuscript less dense:
>
> * We have moved the discussion of interpolation/extrapolation into the appendices. It used to be slightly cumbersome; we have replaced it with Figure 4, which verifies the potential non-robustness of the S4D initialization.
>
> * We have relegated some technical discussion (e.g., Lemma 1) also into the appendices. Instead, we brought a numerical example (see section 3.2) into the main text, which helps elaborate the theory of the diagonal vs DPLR initialization.
>
> * We have made our main practical contribution (perturbing the HiPPO-LegS matrix makes the model more robust) clear in both the introduction and section 4. Moreover, we have added a self-contained discussion about the backward vs forward errors in section 4 in the context of SSMs.
>
> > **Question 1** "Can the author elaborate more on why the proposed method..."
>
> We find two ways of interpreting this question and we will provide our answers to both.
>
> * On one hand, the reviewer may intend to ask why a diagonal structure (or a DPLR structure) of the matrix $\mathbf A$ is used instead of a dense one. In this case, the answer is that a diagonal/DPLR structure of the matrix $\mathbf A$ enables fast evaluation of the model and computation of the gradient. In comparison, evaluating an LTI system with a dense matrix $\mathbf A$ takes $\mathcal{O}(Tn^2)$, where $T$ is the length of the input and $n$ is the size of the state space. This makes training infeasible. We refer interested readers to [Gu et al., 2021] for more information.
>
> * On the other hand, the reviewer may wonder why our S4-PTD (having the diagonal structure) model outperforms the S4 model (having the DPLR structure) on many LRA tasks.  Here are some factors that might be relevant and can possibly explain this advantage of the diagonal structure.
>
>    * The diagonal structure requires fewer parameters to be trained and fewer architectural configurations to be tuned. These possibly make the training easier.
>
>    * A diagonal structure of $\mathbf A$ corresponds to writing the transfer function $G$ into partial fractions while a DPLR structure of $\mathbf A$ corresponds to writing $G$ into the so-called rational barycentric formula. Partial fractions are usually more unstable to parameter perturbations compared to the barycentric formula. In the context of training, this might be an advantage of using a diagonal matrix $\mathbf A$ because it allows us to escape from the local minima even with a small learning rate.
>
>    * This is not an explanation of why the S4-PTD model sometimes outperforms the S4 model, but we want to remark that from an expressiveness perspective, enforcing $\mathbf A$ to be DPLR does not have any advantage over enforcing it to be diagonal, because the transfer function of either $\mathbf A$ is always a rational function with $\leq n$ poles and any such transfer function can be represented using both a diagonal $\mathbf A$ or a DPLR $\mathbf A$. Hence, one should not expect that the diagonal model is naturally inferior to the DPLR model.
>
> We hope this answers the reviewer's questions and concerns. We are happy to answer any follow-up question(s) that the reviewer may have.

---

> > ### Comment · Reviewer_Xrm4 · 2023-11-22
> >
> > I appreciate the author's response.
> > I keep my original score, but agreeing that raising the score to an imaginary 7 is appropriate.

---

> > > ### Author Response · Authors · 2023-11-22
> > > **Thank you for your reply.**
> > >
> > > We thank the reviewer for going through the rebuttal and suggesting that the imaginary 7 would be appropriate.

---

### Author Response · Authors · 2023-11-18
**A list of changes made in the rebuttal revision**

We thank the reviewers for many wonderful comments that helped improve this manuscript a lot. In addition to the minor changes explained in our response to each reviewer, we have compiled a list of major changes (marked in red in the rebuttal revision) made to our manuscript during the rebuttal period:

* We have added a new experiment on the sCIFAR task that shows a strong correlation between the effect of a Fourier-mode perturbation and the location of its frequency relative to the spikes in $|G_{\text{Diag}}|$ (see Figure 4). This verifies our theoretical discussion that the spikes in the transfer function of the S4D initialization make it less robust to certain Fourier-mode perturbations.

* Given our empirical result in Figure 4, we have moved the discussion about the interpolation/extrapolation experiment into the appendices, which deliver the same message in a less obvious way. This improves the cleanliness and conciseness of our manuscript. We have also improved the explanation of Figure 9 (previously as Figure 2) in its caption.

* We have added another new experiment on the sCIFAR task that empirically watches the evolution of the S4D and S4-PTD transfer functions during training (see Figure 13). The experiment shows that empirically, training does not resolve the robustness issue of the S4D model, whereas our S4-PTD model never develops a robustness issue throughout training.

* We have cut a little discussion about our theorems which we believe is slightly too technical. Instead, we have spent more space emphasizing the main practical message that we want to deliver in the introduction: adding a small perturbation to the HiPPO-LegS initialization improves its robustness.

* We have brought a numerical experiment (see Figure 2) that elaborates Theorem 1 to the main text in replacement of Lemma 1. We believe this experiment makes the main message of Theorem 1 more concrete.

* We have added a numerical experiment (see Figure 11) to test the average-case perturbation of the transfer function to accompany Theorem 3, which is about the worst-case perturbation.

* In section 4, we have included a definition of the backward and forward errors in the context of SSMs. This makes the paper more self-contained.

* We have added some supplementary results on the 35-way Speech Command classification task (see Table 6).

* We have added a conclusion to the manuscript, in which we discuss the contribution of the paper and suggest potential future research avenues.

---

### Meta-Review · Area_Chair_6nzT · 2023-12-10

**Metareview:**

The paper introduces a "perturb-then-diagonalize" (PTD) methodology to improve the robustness of SSMs for long-range sequence tasks. This method addresses challenges in diagonalizing the HiPPO framework in S4. The authors propose two models, S4-PTD and S5-PTD, which demonstrate strong convergence to the HiPPO framework and resilience to Fourier-mode noise-perturbed inputs.

Strengths:
- The paper presents a strong theoretical analysis of SSMs and their initialization schemes. The PTD method is well-motivated and grounded in pseudospectral theory.
- The proposed models are empirically validated, showing improved robustness and performance in long-range sequence tasks.
- The approach of addressing diagonalization issues in SSMs using the PTD methodology is novel.
- The paper is well-written, with clear explanations of complex concepts and thorough discussions of the implications of the findings.

Weaknesses:
- Several reviewers noted the paper's complexity and dense writing style, suggesting that some sections could be more accessible.
- Questions were raised about the practical implications of the findings, especially in real-world applications.
- A more extensive comparison with existing models, especially under varied conditions, could strengthen the claims.
- Some reviewers suggested that a deeper exploration of the training dynamics and their impact on model robustness would be beneficial.

Overall, the AC and the reviewers are all in agreement that this is a solid contribution and voted for acceptance.

**Justification For Why Not Higher Score:**

The practical use of the theoretical contributions in real-world applications and their scalability is yet to be explored.

**Justification For Why Not Lower Score:**

The paper makes novel and solid theoretical contributions that would definitely be of the interest of the community as a spotlight paper.

---

### Decision · Program_Chairs · 2024-01-16

Accept (spotlight)